# The role of synoptic circulations in lower-tropospheric dry static energy variability over a South Asian heatwave hotspot

**Hardik M. Shah**[1] **and Joy M. Monteiro**[1,2]

[1]Department of Earth and Climate Science, Indian Institute of Science Education and Research Pune,
Pune, Maharashtra, 411008, India
[2]Department of Data Science, Indian Institute of Science Education and Research Pune, Pune, Maharashtra, 411008, India

**Correspondence:** Hardik M. Shah (hardik.shah.reach@gmail.com)

**Abstract.** TS1 We examine the role of the synoptic-scale circulation in the distribution of daily changes of 600–900 hPa dry static energy (DSE) in a heatwave hotspot in northwest South Asia. Using a combination of linear regression and decision trees, we identify how the quasilinear (mean-eddy) and nonlinear (eddy-eddy) components of the flow contribute to different parts of this distribution. We show that the presence of synoptic eddies leads to strong correlations in the meridional and vertical quasilinear components due to quasigeostrophy, allowing us to identify periods of upper tropospheric eddy activity. We show that the synoptic eddies induce a zonal quasilinear component which plays an important role in governing the magnitude and sign of daily DSE changes. Nonlinear components are observed to play an important role in the tails of this distribution, and we show that the specific nonlinear components that are involved depend on the phase of growth or decay of DSE and of the pre-existing DSE anomaly. We identify energetically distinct configurations involved in the tails of this distribution, and identify eddy configurations corresponding to each of these energetic configurations. Our analysis thus provides a discrete set of "regimes" which can be used to classify extreme daily DSE changes, and provides a more nuanced approach to compositing extreme events which is sensitive to the dynamics underpinning each event.

## 1 Introduction

Recent work has expanded our understanding of the likely processes involved in shaping regional distributions of near-surface temperature (Loikith and Neelin, 2015; Ruff and Neelin, 2012; Petoukhov et al., 2008). Due to the known association of large scale eddies with midlatitude weather extremes (Francis and Vavrus, 2012; Petoukhov et al., 2013), one of the questions of interest has been the role of large-scale dynamics in driving different moments characterizing these distributions (Schneider et al., 2014; Horton et al., 2015; Garfinkel and Harnik, 2016). Given that our ability to understand the impacts of global warming is currently limited by our ability to understand changes in atmospheric circulation (Shepherd, 2014, 2015), understanding the links between circulation and temperature variability in the current climate is important to diagnose the fidelity of climate models in representing these links, leading to improved confidence in climate projections.

Earlier attempts to understand the link between large-scale dynamics and temperature variability used quasilinear models of atmospheric dynamics to address free tropospheric temperature variability, where large-scale dynamics dominate. The choice of quasilinear models was presumably governed by their success in reproducing atmospheric flow statistics (Schneider and Walker, 2006; O'Gorman and Schneider, 2007). However, quasilinear models without nonlinear processes predict Gaussian temperature distributions (Schneider et al., 2014), and are therefore unlikely to provide a reasonable explanation for the observed temperature distributions. While quasilinear models can produce skew by

the incorporation of multiplicative noise processes (Sardeshmukh and Sura, 2009), Garfinkel and Harnik (2016) showed that the inclusion of eddy-eddy interactions can qualitatively reproduce the observed patterns of temperature distributions. There has also been a strong indication that skewness is related to distance from the jetstream via the meridional movement of synoptic systems. We note that the importance of eddy-eddy interactions is not limited to understanding temperature variability, but has been studied in a variety of geophysical contexts (Svirsky et al., 2023; Bouchet et al., 2019; Delsole and Farrell, 1996; Marston and Tobias, 2023). More recent work has tried to build on these initial insights, by using climate models for verifying the importance of nonlinear processes in generating non-Gaussian tails (Linz et al., 2018), and formulating nonlinear approximation schemes (Tamarin-Brodsky et al., 2019).

While such literature has helped expand our understanding of free-tropospheric temperature variability, the widely employed quasilinear models ignore the role played by the zonal mean flow. This choice may have been made due to their interest in the deformation flow field, but this choice makes it problematic to apply their results in an Eulerian framework, which is more relevant for regional temperature variability. Understanding near-surface temperature variability comes with the additional challenge of disentangling the effects of atmospheric macroturbulence and local processes such as land-atmosphere interactions and boundary layer feedback (Miralles et al., 2014; Dirmeyer et al., 2018; Chen and Dirmeyer, 2019). Lately, there have been frameworks proposed for quantifying the contribution of different proximate processes towards driving temperature tendency in different deciles of near-surface temperature (Linz et al., 2020).

Studies which focus on the tails of near-surface temperature distribution have established the association between heatwaves and dynamical processes such as blocking events and quasi-resonant amplification (Petoukhov et al., 2016; Kornhuber et al., 2019; Rao et al., 2021). The synoptic patterns associated with such dynamical processes are usually studied using composites (Ratnam et al., 2016; Rohini et al., 2016). While this strategy is useful for understanding "average" conditions, it can obscure the diversity of pathways that could lead to the outcomes of interest. For example, the composite picture presented in Monteiro and Caballero (2019) while studying extreme wet-bulb temperature events was found to average over at least two mechanisms, which depended on the phase of the wave packet over the region of interest (Pandey et al., 2020).

Since it is unlikely that we will be able to predict the occurrence of extreme events on seasonal timescales, it has been suggested that subseasonal-to-seasonal predictions might benefit from the prediction of waveguides rather than wave driven extremes (White et al., 2021). Furthermore, the relationship between upper tropospheric forcing and near-surface response is not always clear (White et al., 2021), and it might be a useful exercise to understand the impacts of

Rossby wave packets on lower tropospheric quantities that are directly related to near-surface temperature. Such an exercise might allow evaluating seasonal "propensity of extremes" (Prodhomme et al., 2022) directly using such quantities instead. Thus, understanding the variability of such quantities is not only interesting from an atmospheric turbulence viewpoint but also from a seasonal prediction viewpoint. One possible approach (which we take in this study) to addressing the question of understanding near-surface temperature variability and extremes could comprise the following steps:

1. Identifying a lower tropospheric quantity that is highly associated with near-surface temperature and is strongly influenced by the atmospheric circulation

2. Identifying quasilinear and nonlinear contributions to different parts of the distribution of this quantity

3. Characterizing the different pathways to extreme values of this quantity and relating them to characteristics of the circulation

In this study, we identify the lower tropospheric (600–900 hPa) dry static energy $\mathcal{S}_{\mathrm{Tot}}$ (where the subscript "Tot" stands for total), as a suitable quantity whose daily changes are both highly correlated with near-surface temperature and have a significant contribution due to large-scale advection in our region of interest, a heatwave hotspot in South Asia. We study the variability of advection-driven daily changes in $\mathcal{S}_{\mathrm{Tot}}$, during March–April, which are a part of the heatwave season in this region. We study the contribution of quasilinear and nonlinear advective fluxes to the distribution of advection driven daily changes in $\mathcal{S}_{\mathrm{Tot}}$. We use an interpretable machine learning algorithm, the decision tree, to develop a nuanced picture of combinations of advective contributions that lead to extreme daily changes in $\mathcal{S}_{\mathrm{Tot}}$ and show how a combination of these results and compositing leads to a richer description of the different ways in which the upper tropospheric circulation affects daily changes in $\mathcal{S}_{\mathrm{Tot}}$. Finally, we summarize our work with an analysis of the lifecycle of daily changes in $\mathcal{S}_{\mathrm{Tot}}$, highlighting how circulation drives the accumulation, saturation, and ventilation of $\mathcal{S}_{\mathrm{Tot}}$ over the region.

## 2 Data and Methods

### 2.1 Data

We have used the European Center for Medium-Range Weather Forecasts Reanalysis version 5 (ERA5; Hersbach et al., 2020) reanalysis dataset ($0.25° \times 0.25°$ resolution) for analysing daily mean quantities. The period of analysis spans the months of March and April, from 1980 to 2022, with the exclusion of 82 dates due to data quality issues. We use unevenly spaced pressure levels, with higher resolution near the surface and the upper levels. The specific pressure levels (in

hPa) are 100, 125, 150, 175, 200, 225, 250, 300, 400, 500, 600, 700, 750, 775, 800, 825, 850, 875, and 900.

## 2.2 Methodology

Our region of interest is the South Asian heatwave hotspot (Ratnam et al., 2016; Rohini et al., 2016), defined as the area enclosed between 25 and 31° N, and 68 and 78° E. We use dry static energy (DSE) for tracking the energy content of the atmospheric parcels. Since DSE is conserved for dry, large scale adiabatic flows, it is particularly suited for diagnosing energy advection driven by the resolved states of the large scale atmosphere. Using Reynolds decomposition, we decomposed the DSE and velocity fields into daily climatology and anomaly components ($X = \overline{X} + X'$) where the climatology represents the background state and anomalies represent transient variations. Daily climatology was computed as a 10 d rolling mean of daily mean computed over the period of analysis.

Our analysis focuses on the volume of the lower troposphere bounded vertically between 600 and 900 hPa, and horizontally by the spatial extent of the region of interest. As previously mentioned, the dry static energy content of air parcels contained in this volume is given by $\mathcal{S}_{\text{Tot}}$. Daily changes in $\mathcal{S}_{\text{Tot}}$ are given by $\delta \mathcal{S}_{\text{Tot}} := [\mathcal{S}_{\text{Tot}}(t_i) - \mathcal{S}_{\text{Tot}}(t_{i-1})]_{i=1}^{N}$) where $N$ is the total number of days in the analysis. The daily DSE convergence into this volume is calculated as the mass-weighted advection of DSE into it, given by $\delta \mathcal{S}$. The local conservation law for DSE is expressed in Eq. (1), with $\epsilon$ representing the terms involving unresolved or parameterized energy fluxes. Starting with the point conservation law for DSE, TS2

$$\frac{\partial (\text{DSE})}{\partial t} = -\nabla \cdot (\boldsymbol{v}\,\text{DSE}) + \epsilon \tag{1}$$

We analyzed both the divergent and non-divergent components of the RHS; even though the divergent term was not small, advective contributions to the change in DSE were larger than the divergent contributions in the presence of synoptic eddies. The existing literature cited in the introduction (Schneider et al., 2014; Garfinkel and Harnik, 2016; Tamarin-Brodsky et al., 2019; Linz et al., 2020) has also used the advective component for studying the role of eddies in driving midlatitude temperature variability. Further, the ERA5 divergence term may be unreliable due to contamination by numerical noise, as recently acknowledged by in the Copernicus Knowledge Base 2025 (ECMWF, 2025) and recent evaluations (Mayer et al., 2021). Because our objective is not to close the DSE budget but to characterize the role of synoptic eddies in shaping quasilinear and nonlinear DSE flux relationships, we ignore the contribution due to divergence and focus solely on the advective component in this study. Upon neglecting the mass conservation terms and discretizing the quantities, Eq. (1) reduces to the advection form

in Eq. (2).

$$\delta(\text{DSE}) = \delta t \left( -\boldsymbol{v} \cdot \nabla \text{DSE} + \epsilon \right) \tag{2}$$

Taking the mass-weighted integral of Eq. (2), and performing Reynolds decomposition on both sides,

$$\delta \mathcal{S}_{\text{Tot}} = \delta t \left( \oint_D (-\boldsymbol{v} \cdot \nabla \text{DSE} + \epsilon)\, \mathrm{d}M \right)/M \tag{3a}$$

$$\delta \overline{\mathcal{S}}_{\text{Tot}} + \delta \mathcal{S}'_{\text{Tot}} = \delta t \left( \oint_D - \left( v' \frac{\partial \overline{\text{DSE}}}{\partial y} + v' \frac{\partial \text{DSE}'}{\partial y} \right. \right.$$
$$\left. \left. + \overline{v} \frac{\partial \overline{\text{DSE}}}{\partial y} + \overline{v} \frac{\partial \text{DSE}'}{\partial y} + \ldots \right) \mathrm{d}M \right)/M \tag{3b}$$

Using shorthand notation for the convergence terms on the RHS,

$$\delta \overline{\mathcal{S}}_{\text{Tot}} + \delta \mathcal{S}'_{\text{Tot}} = v' \overline{\mathcal{S}}_y + v' \mathcal{S}'_y + \overline{v} \overline{\mathcal{S}}_y + \overline{v} \mathcal{S}'_y + \ldots + \varepsilon \tag{4}$$

Where the order of terms on the RHS is preserved between Eqs. (3b) and (4). Ignoring the small daily changes in $\overline{\mathcal{S}}_{\text{Tot}}$, the mass weighted daily climatology of lower tropospheric DSE, we arrive at the relationship,

$$\delta \mathcal{S}'_{\text{Tot}} = \delta \mathcal{S} + \varepsilon \tag{5}$$

The terms and operations used in Eqs. (1)–(5) are defined as follows:

$$\delta \mathcal{S} = v' \overline{\mathcal{S}}_y + v' \mathcal{S}'_y + \overline{v} \overline{\mathcal{S}}_y + \overline{v} \mathcal{S}'_y + \ldots \tag{6a}$$

$$\boldsymbol{v} = (u, v, w) \tag{6b}$$

$$\oint_D = \int_{Z=Z_{900\,\text{hPa}}}^{Z=Z_{600\,\text{hPa}}} \int_{\lambda=68°}^{\lambda=78°} \int_{\phi=25°}^{\phi=31°} \tag{6c}$$

$$\mathrm{d}M = \rho * (R_e \cos\phi\, \mathrm{d}\lambda) * (R_e\, \mathrm{d}\phi) * (\mathrm{d}Z) \tag{6d}$$

$$M = \oint_D \mathrm{d}M \tag{6e}$$

$$\delta \mathcal{S}'_{\text{Tot}} = \left( \oint_D \delta \text{DSE}'\, \mathrm{d}M \right)/M \tag{6f}$$

$$\overline{\mathcal{S}}_{\text{Tot}} = \left( \oint_D \delta \overline{\text{DSE}}\, \mathrm{d}M \right)/M \tag{6g}$$

where $\delta t = 86\,400$, $\phi =$ Degrees of Latitude, $\lambda =$ Degrees of Longitude, $Z =$ Geopotential, $R_e =$ Radius of the Earth. The integrals above are computed by using discrete spatial steps of length 0.25° in the latitudinal and longitudinal directions. We note that the correlation between $\delta \mathcal{S}_{\text{Tot}}$ and $\delta \mathcal{S}$ was highest when we used 2 d average values of $\delta \mathcal{S}$ given by

$[(\delta\mathcal{S}_{t_i} + \delta\mathcal{S}_{t_{i-1}})/2]_{i=1}^{N}$, where $N$ is the total number of days in the analysis. Consequently, all variables related to advection ($\delta\mathcal{S}$ and its Reynolds components $v'\overline{\mathcal{S}}_y$, $v'\mathcal{S}'_y$, ...) used throughout this study, and the mass-weighted values of velocity and $\mathcal{S}'$ in the advection regimes section (Sect. 5.1), represent 2 d averages.

We use the decision tree classification model, which has been extensively applied in climate science (Gagne et al., 2009; Burrows et al., 1995; Xu et al., 2020; Zhang et al., 2012; Wei et al., 2020). The model recursively partitions the input feature space, generating disjoint branches each corresponding to a set of decision rules that group the samples belonging to the same class together. Each branch consists of internal nodes representing the decision variables or predictors and their optimal split values, and a terminating leaf node representing the dependent variable filtered by the conditions corresponding to its branch. The split values are determined by the criterion of maximum reduction in information entropy (Shannon, 1948). The performance of a decision tree model is usually evaluated using the F1-score (harmonic mean of precision and recall for a given class), averaged over all classes of the dependent (class) variable (Hastie et al., 2001). The model output is visualized as an inverted tree. A detailed description of the decision tree classification model is given in Wei et al. (2020). We analyze the explainability of the identified mechanisms across the distribution of daily advection using linear regression and residual diagnostics.

The entropy associated with a categorical variable $Y$ with $m$ classes is given by

$$H(Y) = -\sum_{i=1}^{m} p_i \log_2(p_i) : P(Y = i) = p_i \tag{7}$$

We first use the decision tree model to identify the conditions primarily associated with different tercile based classes of $\delta\mathcal{S}$. We then analyze the explainability provided by the identified conditions across the distribution of $\delta\mathcal{S}$ using linear regression and residual analysis. Finally, we use the decision tree model for identifying conditions additionally associated with the extreme deciles of $\delta\mathcal{S}$. All calculations are performed using the Python packages xarray (Hoyer and Hamman, 2017), numpy (Harris et al., 2020), pandas (McKinney et al., 2010), statsmodels (Seabold and Perktold, 2010), scikit-learn (Pedregosa et al., 2011), and plotting was done with the help of Python packages matplotlib (Hunter, 2007) and cartopy (Met Office, 2010–2015).

## 3 Lower tropospheric advection links circulations with daily changes in $T_{2\,\mathrm{m}}$ anomaly

Positive $T_{2\,\mathrm{m}}$ anomaly ($T'_{2\,\mathrm{m}}$) is associated with both increased fluxes of shortwave radiation and decreased sensible heat fluxes from the ground (Fig. S1 in the Supplement). In contrast to this, "mega heatwaves" (Domeisen et al., 2023;

Miralles et al., 2014), were associated with increased sensible heat fluxes from dessicated surfaces which are known to contribute to feedback cycles sustaining surface warming over multiple days. On the other hand, we find that daily changes in $T'_{2\,\mathrm{m}}$ (which we denote by $\delta T'_{2\,\mathrm{m}}$) are linked to $\delta\mathcal{S}_{\mathrm{Tot}}$, which in turn are highly correlated with $\delta\mathcal{S}$ (Fig. 1a, b). We find that $\delta T'_{2\,\mathrm{m}}$ is also strongly linked to $\delta\mathcal{S}$, especially at large magnitudes (Fig. 1c). This suggests that the surface fluxes are likely a slower timescale forcing, and that daily variability is primarily governed by the atmospheric flow. Further, we find that the lower tropospheric daily temperature changes are strongly explained by daily $T_{2\,\mathrm{m}}$ changes (Fig. S2). Thus, the daily changes in lower tropospheric DSE capture a large proportion of the variation in the daily changes in $T_{2\,\mathrm{m}}$. Further, March and April are characterized by climatologically low values of precipitation (Fig. S3a) and cloud cover (Fig. S3b). Therefore, we proceed with the hypothesis that daily mean advective fluxes of DSE explain daily mean changes of DSE sufficiently well, and we don't consider contributions from radiative heating and latent heat release in this work. While $\delta\mathcal{S}_{\mathrm{Tot}}$ is unbiasedly approximated by $\delta\mathcal{S}$ in the body of its distribution, parameterized energy fluxes become important for extreme values of $\delta\mathcal{S}$, and cause the systematic overprediction of $\delta\mathcal{S}_{\mathrm{Tot}}$ (Fig. 1b). We note the barotropic nature of $\mathcal{S}'$ TS3 and the eddy wind fields, that strengthen progressively with height (Fig. S4), and that the vertical velocity field is strongly related to quasigeostrophic omega (Fig. S5), highlighting the influence of upper tropospheric dynamics in driving surface anomalies. Motivated by the association between daily changes in near-surface temperature anomaly and advection driven daily changes in lower tropospheric DSE, and the influence of atmospheric circulation, we identify $\delta\mathcal{S}$ as our quantity of interest.

## 4 Primary advective contributions to the $\delta\mathcal{S}$ budget

Our strategy to unravel the relationships governing the variability in advection of lower tropospheric DSE is to first find the underlying mechanisms that govern its sign, and subsequently study the drivers of its magnitude. A description of the magnitudes of the Reynolds components of $\delta\mathcal{S}$ is provided in the Fig. S6. The magnitudes of these components are further justified by analyzing the magnitudes of the climatology and anomaly of $v$ and $\nabla\mathcal{S}$ in the zonal, meridional, and vertical directions (Sect. S2; see also Table S1 and Fig. S7). A zonally oriented climatology of $\mathcal{S}_{\mathrm{Tot}}$ TS4 and horizontal winds is observed in Fig. S8.

This section focuses on the first part of this strategy. $\delta\mathcal{S}$ closely resembles $\mathbb{N}(0, \sigma)$ in its body, with deviation in the tails (Fig. S9). Due to the observed symmetry in the body of $\delta\mathcal{S}$, it is possible to model the separation of its signs by using tercile-based classification of $\delta\mathcal{S}$ with the classes labelled as Negative, Neutral, and Positive, as the dependent variable in a decision tree classification model. The 33rd and 66th

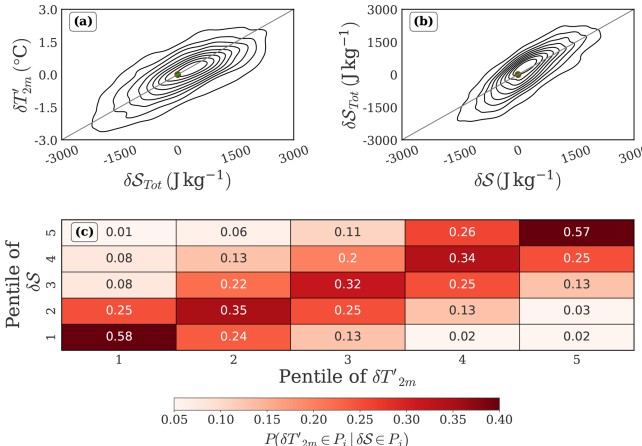

**Figure 1.** This figure shows the pairwise relationships between daily changes in $T_{2m}$ anomaly ($\delta T'_{2m}$), daily changes in 600–900 hPa mass-weighted DSE ($\delta\mathcal{S}_{Tot}$), and daily 600–900 hPa mass-weighted advective convergence of DSE ($\delta\mathcal{S}$) during the combined periods of March and April. **(a)** The joint probability density plot between $\delta T'_{2m}$ and $\delta\mathcal{S}_{Tot}$. **(b)** The joint probability density plot between $\delta\mathcal{S}_{Tot}$ and $\delta\mathcal{S}$. For both these plots, the density contours are estimated using a non-parametric kernel density estimation method; the green dot with the red border is placed at the origin for reference. **(c)** Conditional probability distribution of pentiles of $\delta T'_{2m}$ given the pentile of $\delta\mathcal{S}$. We compute the conditional probability table by cross-tabulating the pentiles of $\delta T'_{2m}$ with the pentiles of $\delta\mathcal{S}$, and then normalizing each row by corresponding row totals to yield $P(\delta T'_{2m} \in P_i | \delta S \in P_j)$ where $P_i$ is the $i$th pentile of $\delta T_{2m}$ and $P_j$ is the $j$th pentile of $\delta\mathcal{S}$. Pentiles refer to five equally populated bins based on ranked values.

percentile values demarcated in Fig. S6a, b. Even though this partitioning seems coarse-grained, we will subsequently show that most large $\delta\mathcal{S}$ events are contained in the Positive and Negative classes defined here.

5    A multivariate decision tree model for tercile-based classes of daily advection retrieves combinations of two of the largest magnitude Reynolds components, namely $\overline{u}\mathcal{S}'_x$ and $w'\overline{\mathcal{S}}_z$ (Fig. 2a). The model is able to identify at least one decision rule for each of the three classes. The decision

10  rules represented by the branches of the tree are described by the conditional probability statements presented in Table 1. The Positive and Negative classes are identified with a larger probability of occurrence than the Neutral class. The decision rules associated with the Positive and Negative classes are

15  based on the in-phase combination of $\overline{u}\mathcal{S}'_x$ and $w'\overline{\mathcal{S}}_z$, while the Neutral class is associated with out-of-phase combinations of $\overline{u}\mathcal{S}'_x$ and $w'\overline{\mathcal{S}}_z$.

The model generated decision rules given in Table 1 are based on a selection of the variables and split points that max-

20  imize entropy reduction (Sect. 2.2). However, the underlying correlation structure of the dataset is not accounted for by the model, specifically, multicollinearity. For example, consider a variable that independently explains a large fraction of the

variance in the dependent variable, but a much smaller frac-
tion when included with one or more of the remaining model    25
predictors. The model algorithm would not identify this vari-
able as important, and its confounded influence would re-
main unidentified. We find that $v'\overline{\mathcal{S}}_y$ is the confounder in our
case, due to a strong inverse association of $v'\overline{\mathcal{S}}_y$ with both
other quasilinear components, $\overline{u}\mathcal{S}'_x$ and $w'\overline{\mathcal{S}}_z$ (Fig. 2b). $v'\overline{\mathcal{S}}_y$    30
is important to decipher the interaction pathways involving
$\overline{u}\mathcal{S}'_x$ and $w'\overline{\mathcal{S}}_z$. Moreover, its magnitude is also comparable to
these variables (Fig. S6c), making its quantitative effect sig-
nificant. Thus, $v'\overline{\mathcal{S}}_y$ is additionally identified as a key com-
ponent in the analysis of $\delta\mathcal{S}$. It will be shown later (Fig. 4a)    35
that the sum of all three quasilinear terms is a strong primary
indicator of advection.

The inverse relationship between $v'\overline{\mathcal{S}}_y$ and $w'\overline{\mathcal{S}}_z$ is traced
back to an in-phase relationship between mass-weighted in-
tegrals of $v'$ and $w'$ over the volume of the box. Due to    40
the vertical coherence of the meridional and vertical velocity
anomaly fields (Fig. S4) and corresponding spatial deriva-
tives of $\overline{\mathcal{S}}$ (Fig. S10), the quasilinear components behave like
linear transformations of the mass-weighted quantities $v'$, $w'$,
which are related through the quasigeostrophic omega equa-    45
tion. The inverse relationship between $v'\overline{\mathcal{S}}_y$ and $\overline{u}\mathcal{S}'_x$ is driven
by the presence of eddies, as will be seen in Sect. 4.1. At the
same time, $w'\overline{\mathcal{S}}_z$ and $\overline{u}\mathcal{S}'_x$ are not strongly correlated, sug-
gesting the role of other mechanisms, either external to the
system or related to different latent relationships with respect    50
to $v'\overline{\mathcal{S}}_y$, governing at least one of the two variables.

## 4.1  Phenomenology

We plot the decision spaces generated by the tree model on
the scatterplot involving all three quasilinear variables iden-
tified above (Fig. 2b). However, it is unclear what physical    55
conditions they represent. Thus, we analyze individual in-
stances and obtain a phenomenology of the large scale spa-
tial patterns of winds and $\mathcal{S}'_{Tot}$ TS5 corresponding to different
regions on the scatterplot. Both periods (March and April)
showed similar results in the following sections; therefore,    60
we focus on presenting an in-depth analysis of April for the
rest of this paper with the corresponding figures for March
presented in the Appendix.

The decision spaces associated with the Negative and Pos-
itive classes have been annotated as Node 1 and Node 4,    65
respectively, along with corresponding probability of occur-
rence in Fig. 2b. The large values of $|v'\overline{\mathcal{S}}_y|$ ($> \sigma$) are asso-
ciated with a highly predictable sign and magnitude of the
other quasilinear quantities, $\overline{u}\mathcal{S}'_x$ and $w'\overline{\mathcal{S}}_z$. Smaller values
($|v'\overline{\mathcal{S}}_y| < \sigma$) are associated with variability in the signs of    70
both other quantities. The extreme deciles (1 and 10) of ad-
vection driven daily changes in $\mathcal{S}$ almost exclusively occu-
pied the decision spaces associated with the Negative and
Positive Nodes 1 and 4 (Figs. 2b, S11), and represented var-
ious configurations of anticyclones and cyclones located in    75
proximity to the region of interest. Figure 3 illustrates how

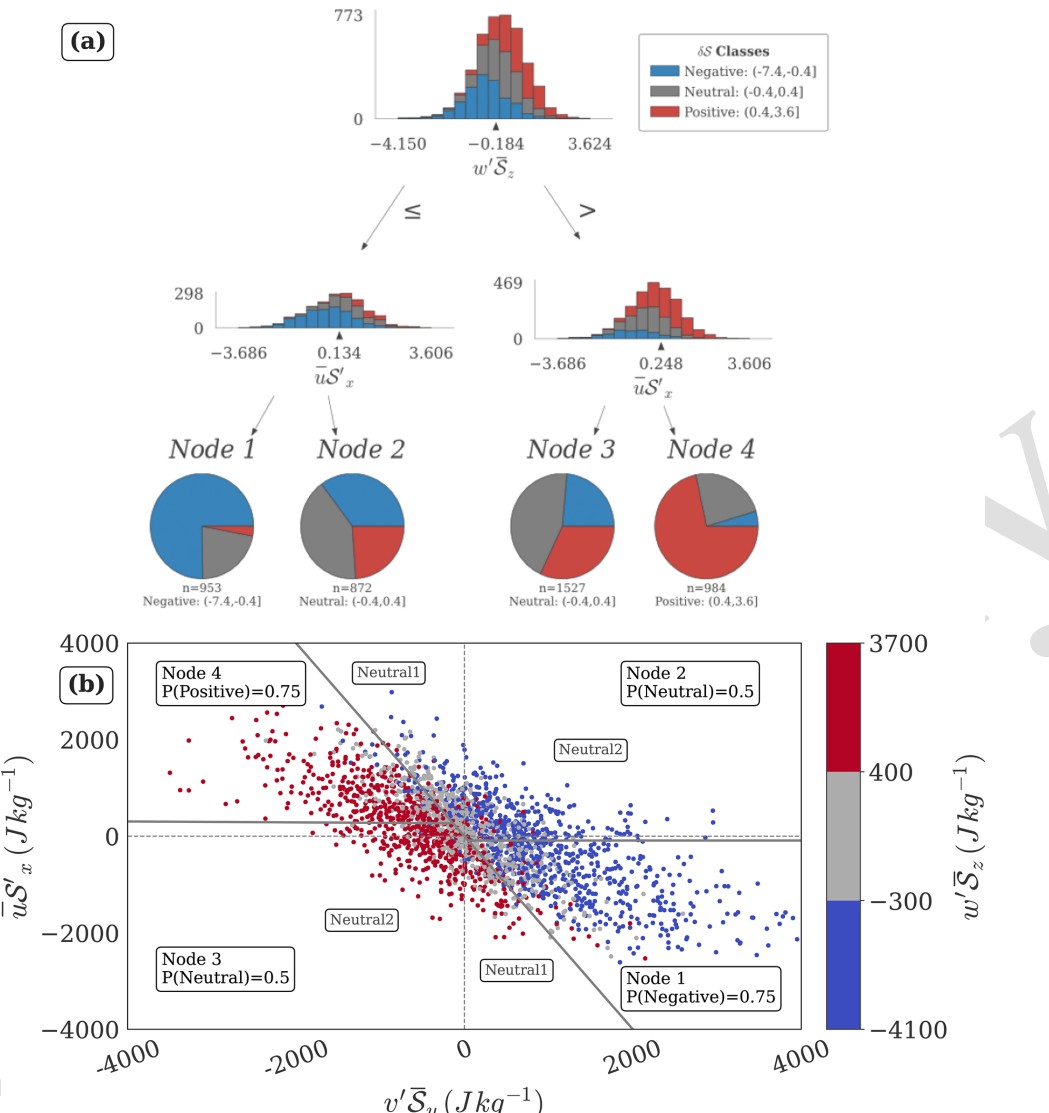

**Figure 2.** This figure examines the relationships governing the differences between terciles of daily advection for the combined period of March and April. **(a)** Standardized decision tree model for tercile based classes of $\delta\mathcal{S}$, across March–April. The root node and the internal (decision) nodes are represented by stacked bar charts of the class frequencies, and the distribution of the classes in the leaf nodes is represented by the pie charts. The variables are standardized, and the split thresholds at each node reflect the number of standard deviations from the mean. The model achieves an F1-score of $\sim 0.6$, with early-stop criteria as follows: entropy decrease $\geq 0.03$ at each level, $\geq 40\,\%$ sample size for level splitting, $\geq 20\,\%$ sample size for leaf nodes. **(b)** Scatterplot showing the relationship between the unstandardized values of $v'\overline{\mathcal{S}}_y$ and $\overline{u}\mathcal{S}'_x$, colored by terciles of $w'\overline{\mathcal{S}}_z$ across March-April. The solid grey lines approximate decision boundaries, with each region labeled by the corresponding Node # and dominant class probability. The labels "Neutral1" and "Neutral2" correspond to the two kinds of deviations from the primary relation described in Sect. 4.1. Pairwise correlations: $r(v'\overline{\mathcal{S}}_y, w'\overline{\mathcal{S}}_z) = -0.75$, $r(v'\overline{\mathcal{S}}_y, \overline{u}\mathcal{S}'_x) = -0.75$, $r(\overline{u}\mathcal{S}'_x, w'\overline{\mathcal{S}}_z) = 0.25$.

the presence of eddies generates the observed phase relationship between $v'\overline{\mathcal{S}}_y$, $\overline{u}\mathcal{S}'_x$ and $w'\overline{\mathcal{S}}_z$.

The decision spaces associated with the Neutral class have been annotated as Node 2 and Node 3 along with probabilities of the Neutral class in Fig. 2b. These decision spaces only weakly prefer the Neutral class with lesser but comparable probabilities of occurrence of the other classes. We find

that these are associated with two kinds of deviations from the expected relationships established above. The first kind is where the meridional and vertical quasilinear components are in phase. We label the regions on the scatterplots representing this kind of deviation as "Neutral1". These points are observed to lie between the lines $\overline{u}\mathcal{S}'_x = -2 * v'\overline{\mathcal{S}}_y$ and $v'\overline{\mathcal{S}}_y = 0$, where $\overline{u}\mathcal{S}'_x * v'\overline{\mathcal{S}}_y < 0$. We find that they represent

**Table 1.** This table describes the decision tree model presented in Fig. 2a. The decision rules and associated class probabilities are approximately the same for both months; therefore, we present the combined results. The means of the involved variables, and their scalar multipliers of $\sigma$, are both $\sim 0$, so expressions such as $0.1\sigma$ are approximated as 0 for ease of readability. For example, $\overline{u}\mathcal{S}'_x \leq -0.1\sigma\,(\overline{u}\mathcal{S}'_x)$ is simplified to $\overline{u}\mathcal{S}'_x \leq 0$. Table headers are described as follows. Advection Class (K): The tercile based classes of $\delta\mathcal{S}$ modeled by the decision tree classification model. Node #: The leaf node numbers identifying each advection class. Conditions ($C$): The set of decision rules that characterize the branches associated with the corresponding leaf node(s) identifying a given advection class. $P(Y = K|C)$: The probability of occurrence of the identified advection class, given the corresponding decision rules.

| Advection Class ($k$) | Node # | Conditions ($C$) | $P(Y = k|C)$ |
|---|---|---|---|
| Negative | 1 | $\overline{u}\mathcal{S}'_x \leq 0,\, w'\overline{\mathcal{S}}_z \leq 0$ | 0.75 |
| Neutral | 2, 3 | $(\overline{u}\mathcal{S}'_x \leq 0,\, w'\overline{\mathcal{S}}_z > 0)\,|\,(\overline{u}\mathcal{S}'_x > 0,\, w'\overline{\mathcal{S}}_z \leq 0)$ | 0.5 |
| Positive | 4 | $\overline{u}\mathcal{S}'_x > 0,\, w'\overline{\mathcal{S}}_z > 0$ | 0.75 |

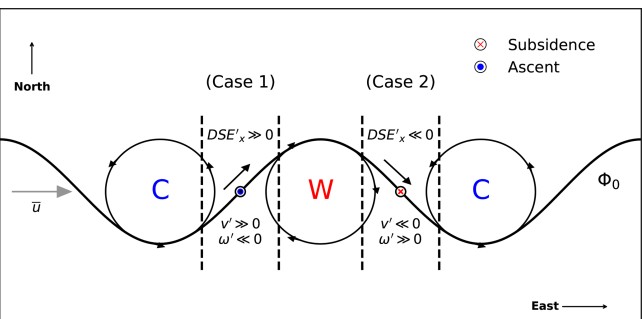

**Figure 3.** This schematic clarifies the relationship between the large scale quantities $v'$, $w'$, and $\mathcal{S}'_x$. Consider a climatological zonal flow ($\overline{u}$) and a geopotential contour of value $\Phi_0$. Let the geopotential contour be displaced meridionally in a sinusoidal manner, with cyclonic and anticyclonic circulations induced along the minima and maxima of the wave, respectively. The corresponding cold and warm anomalies are indicated by letters "C" and "W" respectively. We demarcate the region of interest by a set of dotted vertical lines, and place it first at the head of the cold anomaly (case 1) and then at the head of the warm anomaly (case 2). In case 1, the region experiences a local maximum $v'(\gg 0)$ along with a local minimum $w'(\ll 0)$ both of which advect DSE under a strong climatological DSE gradient in y and z directions; further, since the region here is placed at the edge between the cold and warm anomalies, $\text{DSE}'_x > 0$ is induced and leads to advection of DSE by $\overline{u}$. Such a configuration sees advection components maximize in magnitude with signs as follows: $v'\overline{\mathcal{S}}_y > 0$, $w'\overline{\mathcal{S}}_z < 0$, $\overline{u}\mathcal{S}'_x < 0$. For case 2, all the above signs are reversed and the observed advection components are as follows: $v'\overline{\mathcal{S}}_y < 0$, $w'\overline{\mathcal{S}}_z > 0$, $\overline{u}\mathcal{S}'_x > 0$. Such configurations yield the relationships making up the primary mechanism.

conditions where mass-weighted integrals of $v'$ and $w'$ acted out of phase, i.e., ascending northerly winds and subsiding southerly winds, as expected by the quasigeostrophic relationship. These conditions are characterized by small magnitude ($< \sigma$) of $v'\overline{\mathcal{S}}$, and the horizontal dominance of $\overline{u}\overline{\mathcal{S}}'_x$. Despite the weak meridional winds, instances of large vertical winds are observed in association with mainly the zonal arm of a cyclone or anticyclone located directly to the north or south of the region, when both the meridional arms of an

eddy overlap with the region. Here, $v'$ is rendered small due to averaging over opposite phases, while large $w'$ persists due to homogeneity of its phase across the region, which is in agreement with the phase of quasigeostrophic omega (Fig. S5). $\mathcal{S}'_x$ is purely a function of the differently signed eddies interacting with different zonal halves of the region, and is not sensitive to the $v'$, $w'$ relationship. In cases without eddies, $w'$ is small as $v'$, and only the zonal mean flow advects a large $\mathcal{S}'$ into the region, under a $\mathcal{S}'_x$ governed by no particular structures. Typical configurations associated with the "Neutral1" regions are presented in Fig. S12a, b.

The second kind of deviation is represented by points in the first and third quadrants, where the meridional and zonal quasilinear advection terms are in phase. We label these regions as "Neutral2" in the scatterplot Fig. 2b. The inverse phase relationship between the meridional and vertical quasilinear terms is preserved even at small magnitudes. It represents coherent configurations given by zonally oriented arms of southern/ northern large scale eddies and ridge-like conditions over the region. $\overline{u}\mathcal{S}'_x$ is reversed in sign either because of intensification of $\mathcal{S}'$ along the southern edge of High Mountain Asia, or due to weak larger scale structures of $\mathcal{S}'_x$ not always directly linked with circulation structures (the quantities are 600–900 hPa averaged, and generally, a given location could be influenced by diabatic forcing in the lower vertical levels, or be affected by other proximate eddies). Typical configurations associated with the "Neutral2" regions are presented in Fig. S12c, d.

## 4.2 Explainability and residual bias

We assess the performance of the quasilinear advective contributions by regressing the daily advection of dry static energy into the lower troposphere, $\delta\mathcal{S}$, against the sum of its quasilinear components, $\delta\mathcal{S}_{\text{QL}} = v'\overline{\mathcal{S}}_y + w'\overline{\mathcal{S}}_z + \overline{u}\mathcal{S}'_x$.

Using the coefficient of determination ($R^2$), we find that that $\delta\mathcal{S}_{\text{QL}}$ explains $\sim 65\%$ of the variability in $\delta\mathcal{S}$. To evaluate its performance across the distribution of daily advection, we examine residuals stratified by deciles of $\delta\mathcal{S}$. In the central deciles, the residuals (observation – prediction) are more or less symmetrically distributed about zero (Fig. 4a), indi-

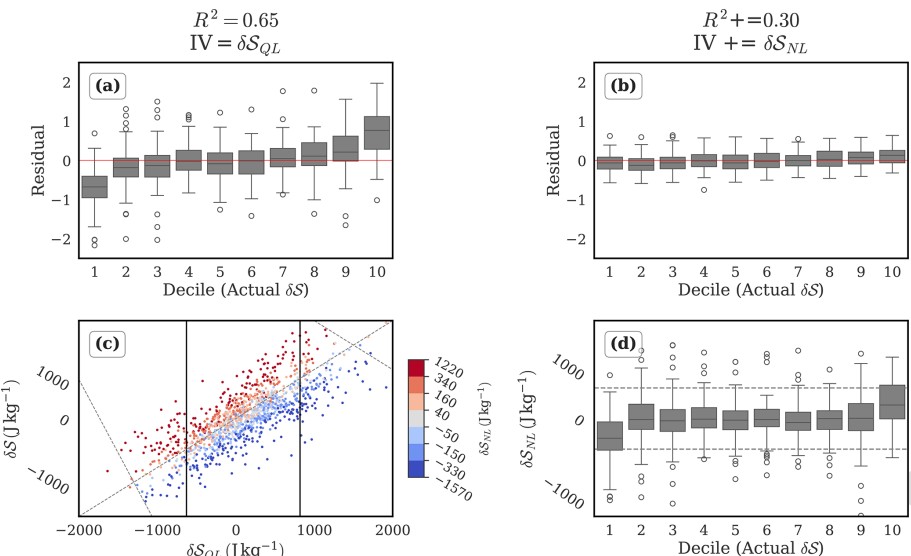

**Figure 4.** This figure presents two regression models for $\delta\mathcal{S}$, and complementary plots based on unstandardized data to aid interpretability, for the period of April. **(a)** Boxplots of residuals grouped by deciles of $\delta\mathcal{S}$, for the fitted standardized regression model $\delta\mathcal{S} \sim 0.8 * \delta\mathcal{S}_{QL} + 0.004$. **(b)** Same as panel **(a)**, but for the fitted standardized regression model $\delta\mathcal{S} \sim 0.95 * \delta\mathcal{S}_{QL} + 0.6 * \delta\mathcal{S}_{NL} + 0.002$. Model performance summaries are provided in Tables S2 and S3. **(c)** Scatterplot between the unstandardized variables, $\delta\mathcal{S}$ and $\delta\mathcal{S}_{QL}$, colored by quantile based categories of $\delta\mathcal{S}_{NL}$. The two negatively sloped dotted lines connect the peak values of $\delta\mathcal{S}$ and $\delta\mathcal{S}_{QL}$ in the right and left tails, highlighting the asymmetric saturation effect of $\delta\mathcal{S}_{NL}$. **(d)** Boxplot of unstandardized values of $\delta\mathcal{S}_{NL}$ grouped by deciles of $\delta\mathcal{S}$.

cating an unbiased model fit. Towards the extremes, however, the residuals exhibit a systematic directional bias (as implied by the increasing magnitude of the mean residuals towards extreme deciles in Fig. 4a), indicating the insufficiency of driving mechanisms in the extremes of the model.

The bias in the primary contributions model is corrected by including the sum of nonlinear variables ($\delta\mathcal{S}_{NL} = v'\mathcal{S}_y' + w'\mathcal{S}_z' + u'\mathcal{S}_x'$). The addition of $\delta\mathcal{S}_{NL}$ eliminates the non-zero mean of residuals in the tails, and reduces the variance of residuals in all the deciles of $\delta\mathcal{S}$, with the largest reduction in the tails (Fig. 4b). The importance of $\delta\mathcal{S}_{NL}$ in the tails of $\delta\mathcal{S}$ is verified by the nonzero mean of $\delta\mathcal{S}_{NL}$ in the extreme deciles of $\delta\mathcal{S}_{QL}$ (Fig. 4d). We infer that the quasilinear contributions are active throughout the distribution of $\delta\mathcal{S}$, and the nonlinear contributions primarily modulate the extremes.

Together, the quasilinear and nonlinear contributions account for $\sim 95\%$ of the variability in $\delta\mathcal{S}$. The rest of the component terms on the RHS of Eq. (6a) account for the remaining $\sim 5\%$ of the variability in $\delta\mathcal{S}$. We shall refer to the quasilinear contributions as the primary contribution, and the nonlinear contributions as the secondary contribution for the remainder of the paper.

Figure 4c shows that the magnitude of $\delta\mathcal{S}$ peaks earlier than $\delta\mathcal{S}_{QL}$ in both tails, suggesting that the main function of the nonlinear terms in the tails is to act against the quasilinear contribution, effectively saturating the growth of $\delta\mathcal{S}$. While this in itself may not seem surprising and has been noted elsewhere (Garfinkel and Harnik, 2016), we also observe a larger difference between the peak values of $\delta\mathcal{S}$ and $\delta\mathcal{S}_{QL}$

in the right tail as compared to the left tail. This asymmetry suggests that the saturative effect of the nonlinear terms is stronger there.

## 5   Secondary advective contributions to the $\delta\mathcal{S}$ budget

The fact that the linear regression model based on $\delta\mathcal{S}_{QL}$ alone causes a skewed distribution of residuals in the extreme deciles of $\delta\mathcal{S}$, and that the residuals are not symmetric about the tails, suggests a crucial role for nonlinearities in determining the asymmetry of the distribution of $\delta\mathcal{S}$. Specifically, the distribution of residuals in the left extreme decile (decile 1) and the right extreme decile (decile 10) is not exactly equal and opposite, highlighting the different influences of $\delta\mathcal{S}_{NL}$ in these regions. While it has been previously suggested that the meridional nonlinear term alone can provide a satisfactory explanation for the observed asymmetry (skewness in their case) (Tamarin-Brodsky et al., 2019), it is unclear whether this is true for our case as well. Thus, the next natural question is to identify the nonlinear components driving the tails of $\delta\mathcal{S}$.

As in the previous section, we use the decision tree model to provide the leading explanatory variables for the extreme deciles 1 and 10 of $\delta\mathcal{S}$. Alongside nonlinear terms, we incorporate the pre-existing anomaly defined as the previous day's total lower tropospheric DSE anomaly, $\mathcal{S}_{\text{Tot, Lag1}}'$, as an explanatory variable. This accounts for the possibility that the drivers of large positive $\delta\mathcal{S}$ may differ depending on

whether advection acts upon a pre-existing positive or negative $\mathcal{S}'_{\text{Tot, Lag1}}$ anomaly. Our results show that not only are distinct nonlinear components active in the left and right tails, but that the influence of these components also varies with the sign of $\mathcal{S}'_{\text{Tot, Lag1}}$. The decision tree model identifies the combinations of variables separating the extreme deciles of $\delta\mathcal{S}$ (Fig. 5a) and leads to the following picture:

– When the *pre-existing anomaly is negative*, ($\mathcal{S}'_{\text{Tot, Lag1}} < 0.51\sigma$; we round the level split thresholds to second decimal) large values of advection – both positive and negative – are associated with the vertical term $w'\mathcal{S}'_z$, as seen in the leftmost branches terminating in Node 1 and Node 2. Positive values of $w'\mathcal{S}'_z > 0.16\sigma$ (extending to $5.38\sigma$) correlate strongly with the positive extremes of $\delta\mathcal{S}$ (Node 2), indicating a strong role of $w'\mathcal{S}'_z$ in dissipating pre-existing negative anomalies. Conversely, small negative values of $w'\mathcal{S}'_z < 0.16\sigma$ (with a very thin tail extending to $-8\sigma$) are associated with Node 1, indicating the potential role of confounding nonlinear variables.

– When the *pre-existing anomaly is positive* ($\mathcal{S}'_{\text{Tot, Lag1}} > 0.51\sigma$), large negative values of $\delta\mathcal{S}$ (Node 3 and Node 4) correspond to strong dissipation of a pre-existing positive anomalies. The model suggests two pathways, associated with *the horizontal terms $v'\mathcal{S}'_y$ and $u'\mathcal{S}'_x$* respectively. $v'\mathcal{S}'_y < -0.24\sigma$ identifies the Negative extreme decile of $\delta\mathcal{S}$ with a 0.85 probability (Node 3). When its values exceed $-0.24\sigma$, but when $u'\mathcal{S}'_x > -0.93\sigma$, there is a a 0.75 probability of occurrence of the Negative extreme decile of $\delta\mathcal{S}$. Large positive values of $\delta\mathcal{S}$ (Node 5) correspond to strong amplification of a pre-existing positive anomaly, and are preferred when $v'\mathcal{S}'_y$ is approximately in its positive half, in combination with small magnitude values of either sign of $u'\mathcal{S}'_x$ (as seen from the histogram of $\delta\mathcal{S}$ at the node where $u'\mathcal{S}'_x$ is split into Nodes 4 and 5).

To contextualize the model results with raw data, we inspect the nonlinear-quasilinear relationships in the context of $\mathcal{S}'_{\text{Tot, Lag1}}$ (Fig. 5b, c, d). First, we see that the nonlinear terms achieve large magnitudes in the extremes of their quasilinear counterparts, corroborating observations from Fig. 4d. Strong, nearly orthogonal relationships emerge in the meridional and vertical directions. Recollecting the strong association between different quasilinear terms and $\delta\mathcal{S}_{\text{QL}}$ (Fig. S13), it is seen that $w'\mathcal{S}'_z > 0$ (first quadrant in Fig. 5d) is strongly associated with dissipation of a pre-existing negative anomaly (since $w'\overline{\mathcal{S}}_z > 0$ corresponds to $\delta\mathcal{S} > 0$). $w'\mathcal{S}'_z < 0$ has fewer instances associated with $\delta\mathcal{S}_{\text{QL}} < 0$, where it causes amplification of a pre-existing negative anomaly. On the other hand, $v'\mathcal{S}'_y$ does not observe as strong a relationship with $\delta\mathcal{S}_{\text{QL}}$ in spite of an apparently weak inverse relationship, when $\mathcal{S}'_{\text{Tot, Lag1}} < 0$. These observations justify the model's choice of $w'\mathcal{S}'_z$ in separating

the extreme deciles of advection when $\mathcal{S}'_{\text{Tot, Lag1}} < 0$. When $\mathcal{S}'_{\text{Tot, Lag1}} > 0$, the horizontal nonlinear-quasilinear relationships seem more active than the vertical relationship, which is reflected in the model output. $v'\mathcal{S}'_y$ seems to weakly prefer being in phase with $\delta\mathcal{S}_{\text{QL}}$, although we see a large frequency of instances with $v'\mathcal{S}'_y > 0$ corresponding to $\delta\mathcal{S}_{\text{QL}} < 0$ conditions. $u'\mathcal{S}'_x$ shows no clear preference except for a strongly negative skew for both signs of pre-existing anomaly. These findings underscore the strong dependency of the sign and magnitudes of the nonlinear terms on their quasilinear counterparts, particularly at the extremes. The schematic in Fig. 6 contextualizes some of the phase relationships observed in Fig. 4a and b.

This analysis highlights the variety of eddy-eddy interactions involved, and the dependence of nonlinear contributions on the quasilinear part of the flow. However, further analysis is needed to establish a direct link between circulation patterns and the relationship between the quasilinear and nonlinear advection components.

## 5.1 Advection regimes

We propose a diagnostic framework to isolate advection regimes by analyzing the phase relationships between primary $\delta\mathcal{S}_{\text{QL}}$ and secondary $\delta\mathcal{S}_{\text{NL}}$ components alongside total advection, conditioned on the sign of pre-existing lower tropospheric DSE anomaly, $\mathcal{S}'_{\text{Tot, Lag1}}$. To this end, we construct a scatterplot of $\delta\mathcal{S}_{\text{QL}}$ versus $\delta\mathcal{S}_{\text{NL}}$, with data points color-coded by deciles of $\delta\mathcal{S}$, allowing us to visualize how combinations of quasilinear and nonlinear contributions map onto the phase and magnitude of net advection. We identify dominant nonlinear drivers for specific regions of the joint distribution, revealing systematic patterns in the interaction between the $\delta\mathcal{S}_{\text{QL}}$ and $\delta\mathcal{S}_{\text{NL}}$ components.

We find that for a given sign of pre-existing anomaly, the extreme deciles of $\delta\mathcal{S}_{\text{QL}}$, when combined with different phases of $\delta\mathcal{S}_{\text{NL}}$, usually correspond to distinct eddy configurations with respect to the region of interest. For instance, pairings of extreme deciles of $\delta\mathcal{S}_{\text{QL}}$ with the central deciles of $\delta\mathcal{S}_{\text{NL}}$ yield different eddy configurations than pairings with its extreme deciles. Pairings with extreme deciles of $\delta\mathcal{S}_{\text{NL}}$ is usually associated with a distinct nonlinear advection term modulating the effect of $\delta\mathcal{S}_{\text{QL}}$, while pairing with central deciles of $\delta\mathcal{S}_{\text{NL}}$ is associated with multiple nonlinear terms that are simultaneously of large magnitude and out of phase, justifying the small magnitude of $\delta\mathcal{S}_{\text{NL}}$. When $\delta\mathcal{S}_{\text{QL}}$ is in its central deciles, fewer pairings with the extreme deciles of $\delta\mathcal{S}_{\text{NL}}$ correspond to coherent eddy configurations. Thus, we identify distinct nonlinear components associated with most eddy configurations where high or low magnitudes of $\delta\mathcal{S}_{\text{NL}}$ interact to amplify or decay the contributions from the extreme deciles of $\delta\mathcal{S}_{\text{QL}}$. We refer to such combinations identified in the phase space as advection regimes.

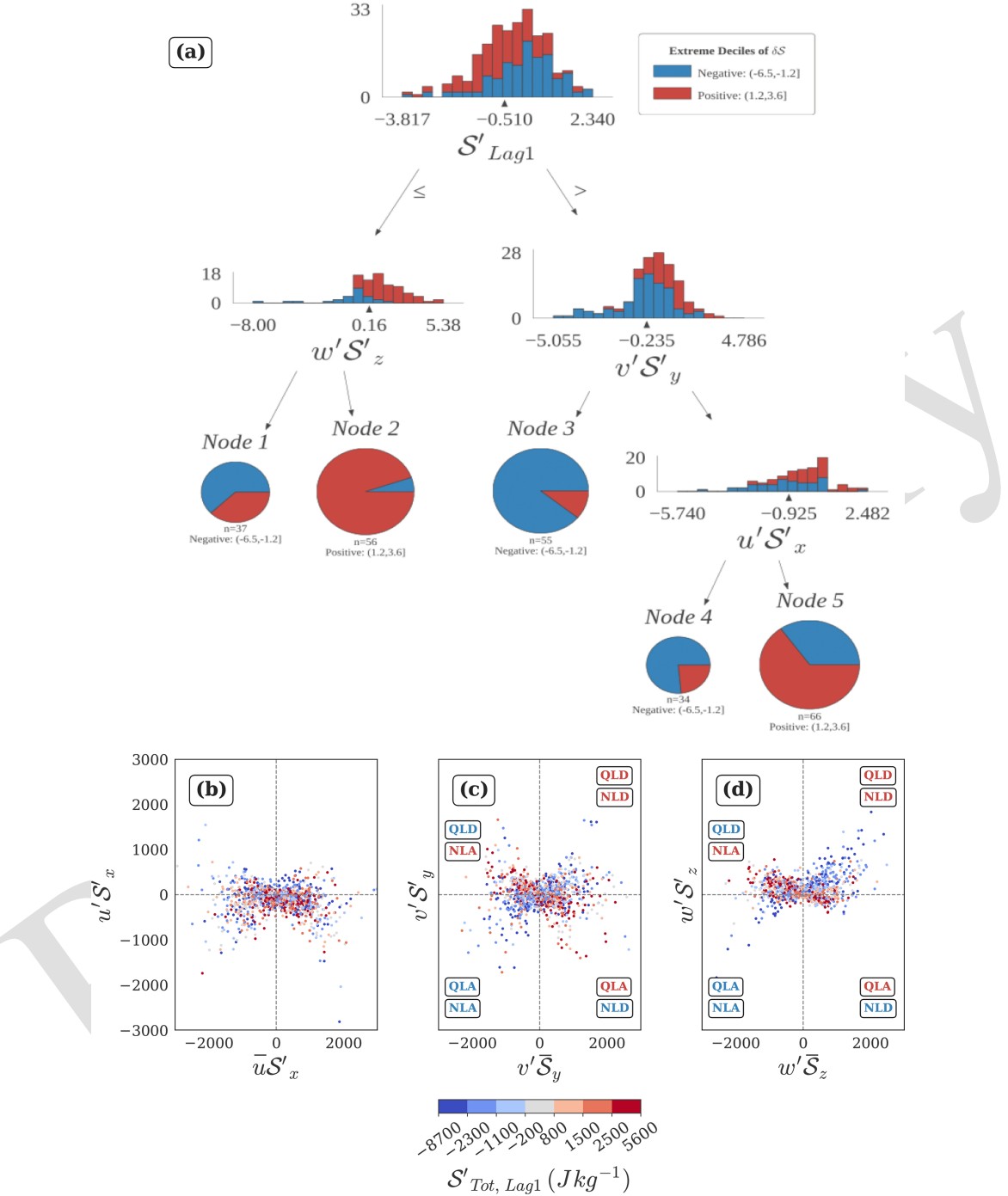

**Figure 5. (a)** Standardized decision tree model identifying the nonlinear drivers of decile-based extremes of $\delta S$ for April. The model achieves an F1-score of $\sim 0.78$, with early-stop criteria as follows: entropy decrease $\geq 0.03$ at each level, $\geq 30\,\%$ sample size for level splitting, $\geq 15\,\%$ sample size for leaf nodes. **(b–d)** Nonlinear-quasilinear relationship in the zonal, meridional, and vertical directions, colored by quantile based categories of $S'_{\text{Tot, Lag1}}$. All the daily mean advection terms are expressed in $\text{J}\,\text{kg}^{-1}$. Quadrants in panels **(c)** and **(d)** are labeled by the role of the advection components in amplifying or decaying the predominant sign of the pre-existing lower tropospheric DSE anomaly ($S'_{\text{Tot, Lag1}}$) in that quadrant, with red for positive and blue for negative. For example, in panel **(c)**, the first quadrant represents cases corresponding to quasilinear decay (QLD) and nonlinear decay (NLD) of a pre-existing negative anomaly by $v'\overline{S}_y > 0$ and $v'S'_y > 0$, respectively. Quadrants where advection components align with the anomaly sign are labeled QLA (Quasilinear Amplification) and NLA (Nonlinear Amplification).

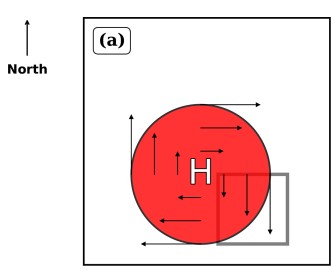
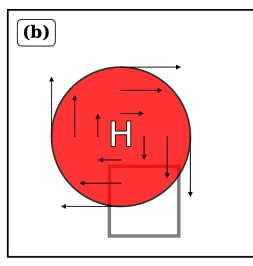

**Figure 6.** This schematic explains the eddy configurations responsible for some of the NL-QL phase interactions observed in Fig. 5b and c. The region of interest is shown by the transparent rectangle with thick grey borders. An anticyclonic warm core eddy associated with a high pressure anomaly is shown by the red circle, and the black solid arrows represent the circulation direction and strength which is proportional to the size of the arrows. The background conditions are given by: $\overline{\mathrm{DSE}}_y < 0$, $\overline{\mathrm{DSE}}_x \sim 0$. **(a)** The second quadrant in Fig. 5c is associated with quasilinear decay and nonlinear amplification of a pre-existing positive DSE anomaly by the meridional advection components. When an anticyclonic eddy is placed beyond the northwest corner of the region, the associated northerly flow ($v' < 0$) advects DSE' into the region due to DSE'$_y > 0$. Such a configuration causes nonlinear amplification of the DSE' of the region. At the same time, the meridional quasilinear advection term acts to dissipate the DSE' with upgradient transport of $\overline{\mathrm{DSE}}$. **(b)** The $\mathcal{S}'_{\mathrm{Tot, Lag1}} > 0$ data points in the fourth quadrant of Fig. 5b are associated with quasilinear amplification and nonlinear decay of a pre-existing positive DSE anomaly by the zonal advection components. When an anticyclonic eddy is placed close to the north of the region, the associated easterly flow ($u' < 0$) advects DSE' out of the region due to DSE'$_x < 0$. Such a configuration causes nonlinear decay of the DSE' of the region, while the zonal quasilinear term acts to amplify the DSE' of the region due to strongly positive ($\overline{u} \gg 0$) mean zonal flow.

### 5.1.1 Pre-existing positive anomaly

For daily cases with a pre-existing negative anomaly ($\mathcal{S}'_{\mathrm{Tot, Lag1}} < 0$), we define Growth by positive daily advective tendency ($\delta\mathcal{S} > 0$) and Decay by negative daily advective tendency ($\delta\mathcal{S} < 0$), reflecting the phase combination of $\delta\mathcal{S}$ relative to the pre-existing positive phase of $\mathcal{S}'_{\mathrm{Tot, Lag1}}$. As established earlier, $\delta\mathcal{S}$ is explained by $\delta\mathcal{S}_{\mathrm{QL}}$ to the first order (Fig. 4a), but in the tails of $\delta\mathcal{S}$, both $\delta\mathcal{S}_{\mathrm{QL}}$ and $\delta\mathcal{S}_{\mathrm{NL}}$ can be comparable in magnitude (Fig. 4c). We study the joint distribution of $\delta\mathcal{S}_{\mathrm{NL}}$ and $\delta\mathcal{S}_{\mathrm{QL}}$, and define Growth regimes for $\mathcal{S}'_{\mathrm{Tot, Lag1}} > 0$ conditioned on $\delta\mathcal{S} > 0$ as follows:

– QL + NL Growth: Nonlinear amplification of Growth, defined by $\delta\mathcal{S}_{\mathrm{NL}} \gg 0$ and $\delta\mathcal{S}_{\mathrm{QL}} \gg 0$, i.e., when both $\delta\mathcal{S}_{\mathrm{NL}}$ and $\delta\mathcal{S}_{\mathrm{QL}}$ are simultaneously in their positive extreme deciles

– NL Growth: Nonlinear Growth, defined by $\delta\mathcal{S}_{\mathrm{NL}} \gg 0$ and $\delta\mathcal{S}_{\mathrm{QL}} \sim 0$, i.e., when $\delta\mathcal{S}_{\mathrm{NL}}$ is in its 10th TS6 decile but $\delta\mathcal{S}_{\mathrm{QL}}$ is in its central deciles (deciles 2 to 9)

– QL Growth: Quasilinear Growth, defined by $\delta\mathcal{S}_{\mathrm{QL}} \gg 0$ and $\delta\mathcal{S}_{\mathrm{NL}} \sim 0$, i.e., when $\delta\mathcal{S}_{\mathrm{QL}}$ is in its 10th decile but $\delta\mathcal{S}_{\mathrm{NL}}$ is in its central deciles (deciles 2 to 9)

– NL Saturated Growth: Nonlinear saturation of Growth, defined by $\delta\mathcal{S}_{\mathrm{NL}} \ll 0$ and $\delta\mathcal{S}_{\mathrm{QL}} \gg 0$, i.e., when $\delta\mathcal{S}_{\mathrm{NL}}$ is in its 1st decile and $\delta\mathcal{S}_{\mathrm{QL}}$ is in its 10th decile

Similarly, we define Decay regimes for $\mathcal{S}'_{\mathrm{Tot, Lag1}} > 0$ conditioned on $\delta\mathcal{S} < 0$ as follows:

– QL + NL Decay: Nonlinear amplification of Decay, defined by $\delta\mathcal{S}_{\mathrm{NL}} \ll 0$ and $\delta\mathcal{S}_{\mathrm{QL}} \ll 0$, i.e., when both $\delta\mathcal{S}_{\mathrm{NL}}$ and $\delta\mathcal{S}_{\mathrm{QL}}$ are simultaneously in their negative extreme deciles

– NL Decay: Nonlinear Decay, defined by $\delta\mathcal{S}_{\mathrm{NL}} \ll 0$ and $\delta\mathcal{S}_{\mathrm{QL}} \sim 0$, i.e., when $\delta\mathcal{S}_{\mathrm{NL}}$ is in its 1st decile but $\delta\mathcal{S}_{\mathrm{QL}}$ is in its central deciles (deciles 2 to 9)

– QL Decay: Quasilinear Decay, defined by $\delta\mathcal{S}_{\mathrm{QL}} \ll 0$ and $\delta\mathcal{S}_{\mathrm{NL}} \sim 0$, i.e., when $\delta\mathcal{S}_{\mathrm{QL}}$ is in its 1st decile but $\delta\mathcal{S}_{\mathrm{NL}}$ is in its central deciles (deciles 2 to 9)

– NL Saturated Decay: Nonlinear saturation of Decay, defined by $\delta\mathcal{S}_{\mathrm{NL}} \gg 0$ and $\delta\mathcal{S}_{\mathrm{QL}} \ll 0$, i.e., when $\delta\mathcal{S}_{\mathrm{NL}}$ is in its 10th decile and $\delta\mathcal{S}_{\mathrm{QL}}$ is in its 1st decile

The above regimes are demarcated in the $\delta\mathcal{S}_{\mathrm{QL}} - \delta\mathcal{S}_{\mathrm{NL}}$ phase space for a pre-existing positive anomaly in Fig. 7, and the identified nonlinear combinations representing these regimes have been annotated in the figure. We observe that $v'\mathcal{S}'_y$ contributes to the nonlinear growth of a pre-existing positive anomaly, while $u'\mathcal{S}'_x$ plays an important role in the nonlinear saturation of its growth. These observations are in agreement with the description in Fig. 6. We also find that both the horizontal nonlinear terms drive nonlinear decay. During QL growth and decay regimes, there can be multiple nonlinear combinations representing large but out of phase nonlinear terms: For example, the QL Decay regime is characterized by two nonlinear combinations of comparable sample size, $u'\mathcal{S}'_x < 0$, $v'\mathcal{S}'_y > 0$ and $u'\mathcal{S}'_x < 0$, $v'\mathcal{S}'_y < 0$, $w'\mathcal{S}'_z > 0$. These findings underscore the variability in nonlinear-quasilinear dynamics and provide a framework for interpreting the evolution of advection structures modifying a pre-existing positive anomaly. The roles of these nonlinear drivers in driving growth and decay of $\mathcal{S}'_{\mathrm{Tot, Lag1}} > 0$ are better clarified by inspecting the nonlinear-quasilinear scatterplots in the context of the phase and magnitude of $\delta\mathcal{S}$, $\delta\mathcal{S}_{\mathrm{QL}}$ and $\delta\mathcal{S}_{\mathrm{NL}}$ (Appendix Figs. A1, A2).

We find that a given advection regime consists of multiple nonlinear combinations. We have annotated the nonlinear combinations with the highest frequencies of occurrence in Fig. 7 – multiple regimes have been annotated if their frequencies are comparably large – and plotted the highest frequency combination in Fig. 8. The dominant nonlinear combinations associated with the advection regimes

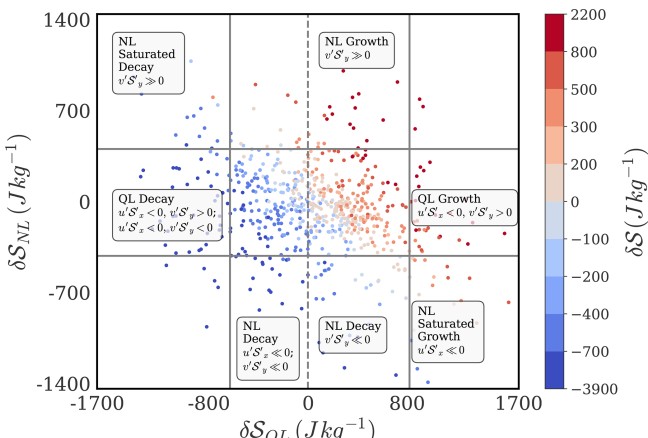

**Figure 7.** This figure shows the phase relationship between $\delta\mathcal{S}_{\mathrm{QL}}$ and $\delta\mathcal{S}_{\mathrm{NL}}$ colored by deciles of $\delta\mathcal{S}$, for pre-existing positive lower tropospheric DSE anomaly ($\mathcal{S}'_{\mathrm{Tot,\,Lag1}} > 0$) conditions during April. Solid grey lines demarcate the 10th and 90th percentiles of $\delta\mathcal{S}_{\mathrm{QL}}$ and $\delta\mathcal{S}_{\mathrm{NL}}$. Regime definitions are provided in Sect. 5.1.1.

for $\mathcal{S}'_{\mathrm{Lag1}} > 0$ conditions are clarified via composite maps (Fig. 8), and described in relation to the spatial configuration of eddy fields below.

1. NL Growth ($v'\mathcal{S}'_y \gg 0$): When the northerly winds associated with either a western/ northwestern anticyclonic disturbance engulf the region, the positive meridional gradient of DSE$'$ intensifies – reaching about half the magnitude of the negative meridional gradient of $\overline{\mathrm{DSE}}$. Simultaneously, the area-averaged northerly $v'$ gains strength and enters its positive extreme decile. Such a configuration enhances the meridional anomalous advection of DSE$'$ into the region, thereby amplifying the effect of positive $\delta\mathcal{S}_{\mathrm{QL}}$ associated with such a configuration.

2. QL Growth ($u'\mathcal{S}'_x < 0$, $v'\mathcal{S}'_y > 0$): When comparable magnitudes of anomalous northerly and easterly winds associated with an anticyclonic disturbance operate in the eastern and southern halves of the region, under comparable magnitudes of $\mathcal{S}'_y > 0$ and $\mathcal{S}'_x < 0$, the easterly winds remove a significant part of the $\mathcal{S}'$ deposited by the northerly winds, rendering $\delta\mathcal{S}_{\mathrm{NL}} \sim 0$. As a result, the net advection is a function of $\delta\mathcal{S}_{\mathrm{QL}}$ alone in this configuration.

3. NL Saturated Growth ($u'\mathcal{S}'_x \ll 0$): When an anticyclone centered to the north of the region has advanced sufficiently into the region, the associated easterly winds pass through the southern half of the region, and the strength of $u' < 0$ increases to more than half of the zonal mean flow. A strongly negative zonal gradient of DSE$'$ is maintained due to the anomalously warm conditions associated with the anticyclone primarily acting on the western boundary of the region. Consequently, $u'$

acts to remove a large part of the DSE$'$ deposited by $\overline{u}$. The magnitude of $v'$ is small in this configuration and does not yield a considerable magnitude of $v'\mathcal{S}'_y$ despite a large magnitude of positive $\mathcal{S}'_y$.

4. NL Decay ($u'\mathcal{S}'_x \ll 0$): When a cyclonic disturbance to the north/west interacts primarily with the western boundary of the (anomalously warm) region of interest, the zonal eddy winds become aligned with the climatological zonal flow. Simultaneously, the zonal gradient of DSE$'$ becomes positive and has a much larger magnitude than its meridional counterpart, because the northern boundary of the region is not affected similarly by the cyclonic disturbance. Coupled with anticyclonic winds to the south of the region, the zonal eddy winds gain strength, and drive the nonlinear decay of the pre-existing positive anomaly of DSE even without a large role of the quasilinear advection component. We note that the NL Decay regime has another component in Fig. 7, characterized solely by $v'\mathcal{S}'_y \gg 0$ within the positive half of $\delta\mathcal{S}_{\mathrm{QL}}$. This subset corresponds to 6 samples exhibiting extreme negative $\delta\mathcal{S}$. Upon inspection of its composite map, we find that it does not represent a coherent eddy configuration. We exclude this component from Fig. 8 and focus our interpretation on the $u'\mathcal{S}'_x \ll 0$ dominated branch of NL Decay described above.

5. NL Saturated Decay ($v'\mathcal{S}'_y \gg 0$): When a cyclone centered to the northwest of the anomalously warm region induces a negative meridional gradient of DSE$'$, strong southerly winds act to deposit DSE$'$ into the region. At the same time, the net quasilinear advection of DSE associated with the cyclone acts to deposit DSE$'$ out of the region. This out of phase relationship is associated with nonlinear saturation of quasilinear decay.

6. QL Decay ($u'\mathcal{S}'_x < 0$, $v'\mathcal{S}'_y > 0$): The QL regime is revisited in the decay phase when a cyclone has advanced sufficiently into the region with anomalous conditions corresponding to strong and opposing zonal and meridional gradients of DSE$'$, and strong westerly and southerly anomalous winds. In such a configuration, the nonlinear advection terms cancel out, rendering $\delta\mathcal{S}_{\mathrm{NL}}$ of low magnitude.

We study the conditional probability distribution of deciles of $\mathcal{S}'_{\mathrm{Tot,\,Lag1}}$ given the occurrence of the advection regimes active during $\mathcal{S}'_{\mathrm{Tot,\,Lag1}} > 0$ conditions (Table 2). NL Growth and QL Growth regimes are predominantly concentrated in the central deciles of $\mathcal{S}'_{\mathrm{Tot,\,Lag1}} > 0$, suggesting that linear and nonlinear growth mechanisms are most active during moderate positive anomalies. In contrast, the NL Saturated Growth regime preferentially occurs during more extreme positive values of $\mathcal{S}'_{\mathrm{Tot,\,Lag1}}$. NL Decay and QL Decay regimes are most active during moderately positive $\mathcal{S}'_{\mathrm{Tot,\,Lag1}}$ conditions, and NL Saturated Decay during neutral $\mathcal{S}'_{\mathrm{Tot,\,Lag1}}$ conditions.

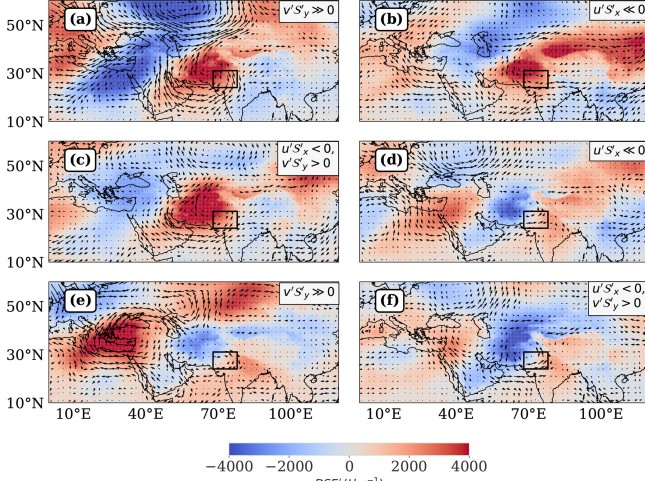

**Figure 8.** This figure shows the composite representations of horizontal eddy wind vectors and $\mathcal{S}'_{\text{Tot}}$ corresponding to the dominant nonlinear combination per advection regime amplifying and dissipating a pre-existing positive lower tropospheric DSE anomaly ($\mathcal{S}'_{\text{Tot, Lag1}} > 0$) during April, as defined in Fig. 7. The order of regimes is as follows: **(a)** NL Growth, **(b)** NL Saturated Growth, **(c)** QL Growth, **(d)** NL Decay, **(e)** NL Saturated Decay, **(f)** QL Decay. The upper right textbox in each panel annotates the dominant nonlinear combination for the corresponding regime. The Table S4 clarifies the representativeness of an advection regime by the plotted nonlinear combination in the column "% Contribution". The color coding represents $\mathcal{S}'_{\text{Tot}}$ values ranging from $-4000$ to $4000$ J kg$^{-1}$, with stronger hues indicating larger magnitudes.

**Table 2.** This table shows the conditional probability distribution of ranked deciles of $\mathcal{S}'_{\text{Tot, Lag1}}$ given the occurrence of an advection regime defined for $\mathcal{S}'_{\text{Lag1}} > 0$ TS7 conditions during April. We compute the conditional probability table by cross-tabulating the deciles of $\mathcal{S}'_{\text{Tot, Lag1}}$ with the advection regimes identified for $\mathcal{S}'_{\text{Lag1}} > 0$ conditions in Fig. 8, and then normalizing each row by corresponding row totals. Each cell value represents the conditional probability $P(\mathcal{S}'_{\text{Lag1}} \in D_i | \text{regime} \in A_j)$ where $D_i$ is the $i$th decile of $\mathcal{S}'_{\text{Tot, Lag1}}$ and $A_j$ is the advection regime on the $j$th row.

| Advection Regime | 5 | 6 | 7 | 8 | 9 | 10 |
|---|---|---|---|---|---|---|
| NL Growth | 0.19 | 0.25 | 0.19 | 0.12 | 0.19 | 0.06 |
| NL Saturated Growth | 0.13 | 0.20 | 0.13 | 0.00 | 0.07 | 0.47 |
| QL Growth | 0.11 | 0.22 | 0.11 | 0.33 | 0.11 | 0.11 |
| NL Decay | 0.14 | 0.10 | 0.19 | 0.19 | 0.24 | 0.14 |
| NL Saturated Decay | 0.00 | 0.54 | 0.15 | 0.15 | 0.15 | 0.00 |
| QL Decay | 0.06 | 0.25 | 0.28 | 0.16 | 0.09 | 0.16 |

Overall, most of the advection regimes are distributed across neutral, moderate and extreme positive deciles of $\mathcal{S}'_{\text{Tot, Lag1}}$, indicating that large Growth and Decay processes remain active throughout the lifecycle of $\mathcal{S}'_{\text{Tot, Lag1}} > 0$.

### 5.1.2 Pre-existing negative anomaly

For daily cases with a pre-existing negative anomaly ($\mathcal{S}'_{\text{Tot, Lag1}} < 0$), we define Growth by negative daily advective tendency ($\delta\mathcal{S} < 0$) and Decay by positive daily advective tendency ($\delta\mathcal{S} > 0$), reflecting the phase combination of $\delta\mathcal{S}$ relative to the pre-existing negative phase of $\mathcal{S}'_{\text{Tot, Lag1}}$. We study the joint distribution of $\delta\mathcal{S}_{\text{NL}}$ and $\delta\mathcal{S}_{\text{QL}}$, and define Growth regimes for $\mathcal{S}'_{\text{Tot, Lag1}} < 0$ conditioned on $\delta\mathcal{S} < 0$ as follows:

- QL + NL Growth: Nonlinear amplification of Growth, defined by $\delta\mathcal{S}_{\text{NL}} \ll 0$ and $\delta\mathcal{S}_{\text{QL}} \ll 0$, i.e., when both $\delta\mathcal{S}_{\text{NL}}$ and $\delta\mathcal{S}_{\text{QL}}$ are simultaneously in their negative extreme deciles

- NL Growth: Nonlinear Growth, defined by $\delta\mathcal{S}_{\text{NL}} \ll 0$ and $\delta\mathcal{S}_{\text{QL}} \sim 0$, i.e., when $\delta\mathcal{S}_{\text{NL}}$ is in its 1st decile but $\delta\mathcal{S}_{\text{QL}}$ is in its central deciles (deciles 2 to 9)

- QL Growth: Quasilinear Growth, defined by $\delta\mathcal{S}_{\text{QL}} \ll 0$ and $\delta\mathcal{S}_{\text{NL}} \sim 0$, i.e., when $\delta\mathcal{S}_{\text{QL}}$ is in its 1st decile but $\delta\mathcal{S}_{\text{NL}}$ is in its central deciles (deciles 2 to 9)

- NL Saturated Growth: Nonlinear saturation of Growth, defined by $\delta\mathcal{S}_{\text{NL}} \gg 0$ and $\delta\mathcal{S}_{\text{QL}} \ll 0$, i.e., when $\delta\mathcal{S}_{\text{NL}}$ is in its 10th decile and $\delta\mathcal{S}_{\text{QL}}$ is in its 1st decile

Similarly, we define Decay regimes for $\mathcal{S}'_{\text{Tot, Lag1}} < 0$ conditioned on $\delta\mathcal{S} > 0$ as follows:

- QL + NL Decay: Nonlinear amplification of Decay, defined by $\delta\mathcal{S}_{\text{NL}} \gg 0$ and $\delta\mathcal{S}_{\text{QL}} \gg 0$, i.e., when both $\delta\mathcal{S}_{\text{NL}}$ and $\delta\mathcal{S}_{\text{QL}}$ are simultaneously in their positive extreme deciles

- NL Decay: Nonlinear Decay, defined by $\delta\mathcal{S}_{\text{NL}} \gg 0$ and $\delta\mathcal{S}_{\text{QL}} \sim 0$, i.e., when $\delta\mathcal{S}_{\text{NL}}$ is in its 10th decile but $\delta\mathcal{S}_{\text{QL}}$ is in its central deciles (deciles 2 to 9)

- QL Decay: Quasilinear Decay, defined by $\delta\mathcal{S}_{\text{QL}} \gg 0$ and $\delta\mathcal{S}_{\text{NL}} \sim 0$, i.e., when $\delta\mathcal{S}_{\text{QL}}$ is in its 10th decile but $\delta\mathcal{S}_{\text{NL}}$ is in its central deciles (deciles 2 to 9)

- NL Saturated Decay: Nonlinear saturation of Decay, defined by $\delta\mathcal{S}_{\text{NL}} \ll 0$ and $\delta\mathcal{S}_{\text{QL}} \gg 0$, i.e., when $\delta\mathcal{S}_{\text{NL}}$ is in its 1st decile and $\delta\mathcal{S}_{\text{QL}}$ is in its 10th decile

The above regimes are demarcated in the $\delta\mathcal{S}_{\text{QL}} - \delta\mathcal{S}_{\text{NL}}$ phase space for a pre-existing negative anomaly in Fig. 9, and the identified nonlinear combinations representing these regimes have been annotated in the figure. When $\mathcal{S}'_{\text{Tot, Lag1}} < 0$, $u'\mathcal{S}'_x$ contributes to nonlinear amplification of growth, and $v'\mathcal{S}'_y$ aids nonlinear saturation of the growth of the negative anomaly, as observed in Fig. 9. In the decay phase, the roles of the nonlinear terms are reversed, and $u'\mathcal{S}'_x$ drives nonlinear saturation of decay while $v'\mathcal{S}'_y$ drives nonlinear amplification of decay. We see different nonlinear combinations active

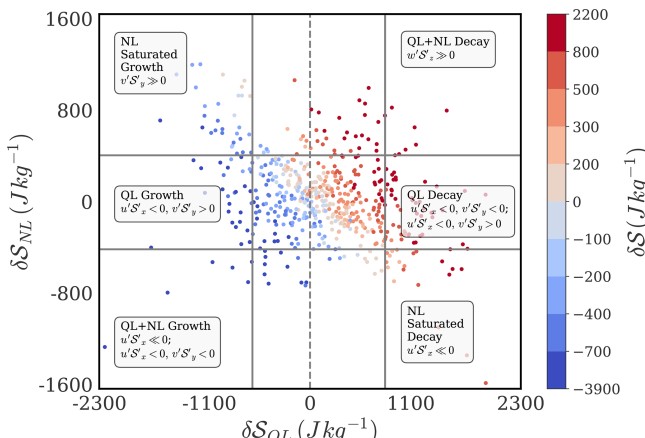

**Figure 9.** This figure shows the phase relationship between $\delta\mathcal{S}_{\mathrm{QL}}$ and $\delta\mathcal{S}_{\mathrm{NL}}$ colored by deciles of $\delta\mathcal{S}$, for pre-existing negative lower tropospheric DSE anomaly ($\mathcal{S}'_{\mathrm{Tot,\,Lag1}} < 0$) conditions during April. Solid grey lines demarcate the 10th and 90th percentiles of $\delta\mathcal{S}_{\mathrm{QL}}$ and $\delta\mathcal{S}_{\mathrm{NL}}$. Regime definitions are provided in Sect. 5.1.2.

in the QL Decay regime; e.g. $u'\mathcal{S}'_x < 0$, $v'\mathcal{S}'_y < 0$, $w'\mathcal{S}'_z > 0$ and $u'\mathcal{S}'_x < 0$, $v'\mathcal{S}'_y > 0$. As before, the roles of these non-linear terms in driving growth and decay of $\mathcal{S}'_{\mathrm{Tot,\,Lag1}} < 0$ conditions are better understood by inspecting nonlinear-quasilinear scatterplots in the context of the phase and magnitude of $\delta\mathcal{S}$, $\delta\mathcal{S}_{\mathrm{QL}}$ and $\delta\mathcal{S}_{\mathrm{NL}}$ (Appendix Figs. A3, A4).

The circulations associated with the dominant nonlinear combinations representing the advection regimes for $\mathcal{S}'_{\mathrm{Lag1}} < 0$ conditions (Fig. 9) are clarified via composite maps of eddy wind and DSE fields (Fig. 10), and described in relation to the spatial configuration of eddy fields below.

1. QL + NL Growth ($u'\mathcal{S}'_x \ll 0$): When a cyclonic disturbance is located to the northwest of the region, predominantly its easterly winds advect cold DSE into the region. This effect is intensified when the easterly flow is reinforced by easterly jet streaks, resulting in nonlinear amplification of the growth of a pre-existing negative DSE anomaly.

2. NL Saturated Growth ($v'\mathcal{S}'_y \ll 0$ TS8): When a cyclonic disturbance centered to the west induces a negative meridional gradient of DSE', the eddy winds over the region are dominated by southerly flow that advects DSE' into the region. In this configuration, positive nonlinear advection in the meridional direction acts in opposite phase to the negative quasilinear advection, leading to nonlinear saturation of the growth of the negative DSE anomaly.

3. QL Growth ($u'\mathcal{S}'_x < 0$, $v'\mathcal{S}'_y > 0$): When comparable magnitudes of eddy southerlies and westerlies associated with a cyclonic disturbance prevail in the eastern and southern halves of the region respectively, and coincide with similarly strong but opposite signed gradients

of DSE' in respective directions, the southerly winds remove a substantial portion of the colder DSE' deposited by the westerly winds. This mutual cancellation of opposing contributions renders $\delta\mathcal{S}_{\mathrm{NL}} \sim 0$. As a result, the net advection is governed solely by $\delta\mathcal{S}_{\mathrm{QL}}$ in this configuration.

4. QL + NL Decay ($w'\mathcal{S}'_z \gg 0$): When northerly anticyclonic eddy winds are strongly coupled with subsiding vertical eddy winds, acting on a negative meridional and positive vertical gradient of DSE', the meridional nonlinear advection term anomalously cools the region, while the vertical nonlinear advection term anomalously warms it – with the latter exerting a dominant effect. The vertical nonlinear advection term acts in phase with net quasilinear advection, resulting in nonlinear amplification of the decay of a pre-existing negative DSE anomaly.

5. NL Saturated Decay ($u'\mathcal{S}'_x \ll 0$): We find a configuration representative of a trough-ridge couplet, with the southward dipping arm of cyclonically curved easterly winds of the ridge sweeping across the region. Coupled with a warm trough to the northwest of the region, the easterly arm advects cold DSE' into the region, while the northerly winds are associated with positive quasilinear advection of DSE'. Thus, the zonal nonlinear advection acts in the opposite phase of the quasilinear advection associated with such a configuration.

6. QL Decay ($u'\mathcal{S}'_x < 0$, $v'\mathcal{S}'_y < 0$, $w'\mathcal{S}'_z \gg 0$): The QL regime is revisited in the decay phase when northeasterly winds advect a southeastward horizontal gradient of DSE' to amplify the pre-existing negative DSE', but are opposed by large and positive nonlinear advection in the vertical, rendering $\delta\mathcal{S}_{\mathrm{NL}}$ of low magnitude.

Thus, for a negative pre-existing anomaly, the roles of the meridional and zonal nonlinear terms were strongly reversed in contributions to amplification and dissipation. $v'\mathcal{S}'_y$ was out of phase with $\delta\mathcal{S}$ during both growth and decay since $\mathcal{S}'_y$ was of the same sign as $\overline{\mathcal{S}}_y$, and was out of phase with $\delta\mathcal{S}_{\mathrm{QL}}$. $u'\mathcal{S}'_x$ strongly aided the amplification of the negative pre-existing anomaly since the easterly winds associated with cyclonic winds lined up with the climatological zonal winds, which were nearly always easterly. $w'\mathcal{S}'_z$ made significant contributions to the amplification phase.

We examine the conditional probability distribution of the deciles of $\mathcal{S}'_{\mathrm{Tot,\,Lag1}}$ given the occurrence of the advection regimes active during $\mathcal{S}'_{\mathrm{Tot,\,Lag1}} < 0$ conditions (Table 3). QL + NL Growth exhibits a strong preference for the left tail of $\mathcal{S}'_{\mathrm{Tot,\,Lag1}}$, nonlinearly amplifying the growth of pre-existing extreme negative values of $\mathcal{S}'_{\mathrm{Tot,\,Lag1}}$. NL Saturated Growth spans both extreme and moderate negative values of $\mathcal{S}'_{\mathrm{Tot,\,Lag1}} < 0$, while QL Growth is most active during moderate values. QL + NL Decay and QL Decay are strongly

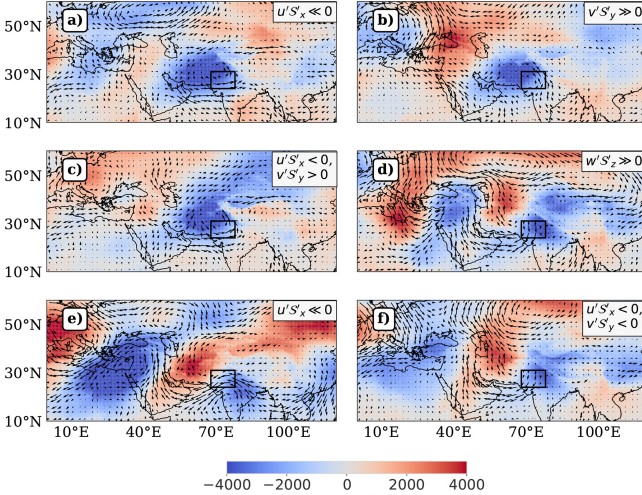

**Figure 10.** This figure shows the composite representations of horizontal wind anomaly vectors and $\mathcal{S}'_{\text{Tot, Lag1}}$ corresponding to the dominant nonlinear combination per advection regime amplifying and dissipating a pre-existing negative lower tropospheric DSE anomaly ($\mathcal{S}'_{\text{Tot, Lag1}} < 0$) during April, as defined in Fig. 9. The order of regimes is as follows: **(a)** QL + NL Growth, **(b)** NL Saturated Growth, **(c)** QL Growth, **(d)** QL + NL Decay, **(e)** NL Saturated Decay, **(f)** QL Decay. The upper right textbox in each panel annotates the dominant nonlinear combination for the corresponding regime. Table S5 clarifies the representativeness of an advection regime by the plotted nonlinear combination in the column "% Contribution". The color coding represents $\mathcal{S}'_{\text{Tot}}$ values ranging from $-4000$ to $4000 \, \text{J kg}^{-1}$, with stronger hues indicating larger magnitudes.

**Table 3.** This table shows the same conditional probability distribution defined in Table 2, but filtered for cases with a negative pre-existing anomaly (TS9 $\mathcal{S}'_{\text{Lag1}} < 0$) during April. We compute the conditional probability table by cross-tabulating the deciles of $\mathcal{S}'_{\text{Tot, Lag1}}$ with the advection regimes identified for $\mathcal{S}'_{\text{Lag1}} < 0$ conditions in Fig. 10, and then normalizing each row by corresponding row totals.

| Advection Regime | 1 | 2 | 3 | 4 | 5 |
|---|---|---|---|---|---|
| QL + NL Growth | 0.33 | 0.27 | 0.07 | 0.20 | 0.13 |
| NL Saturated Growth | 0.15 | 0.37 | 0.26 | 0.30 | 0.09 |
| QL Growth | 0.17 | 0.17 | 0.26 | 0.30 | 0.09 |
| QL + NL Decay | 0.71 | 0.00 | 0.29 | 0.00 | 0.00 |
| NL Saturated Decay | 0.14 | 0.29 | 0.24 | 0.10 | 0.24 |
| QL Decay | 0.34 | 0.28 | 0.19 | 0.16 | 0.03 |

concentrated in the left tail of $\mathcal{S}'_{\text{Tot, Lag1}}$, and NL Saturated Decay is more evenly distributed across the negative range of $\mathcal{S}'_{\text{Tot, Lag1}}$. Notably, all the Growth and Decay regimes here preferentially occupy moderate and extreme deciles of $\mathcal{S}'_{\text{Tot, Lag1}} < 0$ – a contrast to the regime behavior observed for $\mathcal{S}'_{\text{Tot, Lag1}} > 0$ conditions in Sect. 5.1.1.

We repeat the entire analysis for the period of March, and arrive at distinct advection regimes associated with it (Appendix Figs. B1, B2). Overall, there are indications that such mapping between circulation patterns and the NL-QL phase space can effectively map different phases of the transition of a large scale disturbance over the region of interest.

## 6 Summary and Discussion

In this study, we have aimed to identify the contribution of atmospheric circulation to daily change in daily mean lower tropospheric DSE during March-April in a subtropical heatwave hotspot in South Asia. We choose to study DSE changes because we saw that while the daily temperature field is best explained by surface and radiative processes, *daily changes* in temperature are better explained by advection of DSE in the lower troposphere. Thus, understanding daily lower tropospheric DSE changes helps understand the variability in daily temperature changes over the region, particularly when the change is large.

We find that the velocity and DSE anomaly fields are vertically coherent, so the choice of this quantity allows us to focus on the impacts of upper tropospheric forcing on the lower tropospheric DSE budget, regardless of the form of the forcing itself (wave propagation, breaking, etc.). Forcing due to balanced dynamics manifests itself as correlations between the different quasilinear advective terms, whereas such correlations may be absent for other kinds of circulations (such as monsoon intraseasonal oscillations) which may be more prevalent during or after the onset of the monsoon in this region.

We use decision trees to identify primary and secondary drivers of DSE variability because decision trees are both explainable and provide a relative quantification of the contribution of individual physical processes to DSE variability. Each model output is supported with an analysis of the raw data, ensuring the decision rules are both interpretable and grounded in robust underlying relationships.

We find that the "primary" quasilinear relationship capturing the inverse relationships between the meridional and vertical quasilinear components, and the meridional and zonal quasilinear components of advection effectively separates the presence of eddies from incoherent flows, especially at large magnitudes. Smaller magnitudes of advection can represent eddy structures even when these relationships are not obeyed, but most instances with small magnitudes represent the absence of eddies.

Further, we find that the events that constitute the tail of the distribution of $\delta\mathcal{S}$ represent a rich variety of eddy configurations or "advection regimes" that are missing in the literature of temperature variability. We also find a strong presence of quasilinear regimes in the tails of $\delta\mathcal{S}$, and, counter to expectation, the presence of nonlinear regimes in the body of $\delta\mathcal{S}$. We obtain a circulation map for some of these regimes that

are satisfactorily explained in terms of the horizontal configuration of the eddies.

The following list summarises the important observations and takeaways from our analysis:

1. We find that the quasilinear advective terms are important across all terciles of daily advective DSE changes. In contrast, the nonlinear terms are important primarily in the tails of the distribution. This observation is in line with previous studies, which propose that atmospheric macroturbulence is predominantly quasilinear – characterised by weak eddy-eddy interactions – and that eddy-eddy interactions are important for explaining the tail behavior of such macroturbulence. Further, the particular eddy-eddy interactions active in the tails of the distribution differ between extreme positive and negative advective DSE changes, depending on the sign of the pre-existing DSE anomaly.

2. We see that the combined vertical and zonal advection of DSE overwhelms the meridional advection of DSE, leading to an interesting situation where *southward* advection leads to an increase in lower tropospheric DSE, opposite to what one would expect if temperature variability was considered as a problem of one-dimensional mixing of the climatological equator-pole temperature gradient by eddies.

3. In the presence of eddies, the quasigeostrophically driven meridional-vertical wind coupling becomes associated with a zonal gradient of DSE′ across the region of interest that persists during the transition of an eddy due to a preferred zonal direction of propagation (Fig. 3). The importance of vertical advection may be linked to the large static stability in our region of interest. The advection of DSE′ by zonal mean flow is comparable in magnitude to the primary meridional and vertical terms due to large zonal flows prevalent during the considered times of the year. Thus, the zonal term has decisive control over the net effect of the quasilinear coupling involving all these three terms. This contribution appears to have been ignored in previous analyzes due to the focus on advection of isotherms by eddy flow fields, ignoring the zonal mean flow inhomogeneities between regions of a given latitude.

4. In addition to the large eddy winds in the meridional and vertical directions and a large zonal gradient of DSE′ in the presence of an eddy, different eddy configurations render one or more of the following large: meridional gradient of DSE′, vertical gradient of DSE′ and zonal eddy winds ($u'$). Since the nonlinear advection terms result from the product of these terms in respective directions, different nonlinear terms become large under different eddy configurations and become important in the tails of the distribution of $\delta\mathcal{S}$. We note that eddies

can also exist in the body of the distribution of daily advection of DSE, but we do not analyze them separately since they represent small advective daily changes in DSE.

5. $u'\mathcal{S}'_x$ is also neglected in studies considering nonlinear advection processes (Tamarin-Brodsky et al., 2019), presumably due to the Lagrangian approach which is not affected by the secondary gradients induced across a region. We find that zonal nonlinear advection term is mostly negative, irrespective of the sign of the pre-existing anomaly. This is because the signs of the anomalous zonal velocity and $DSE'_x$ are opposite in the case of a northwestern anticyclone as well as a northwestern cyclone. The role of $u'\mathcal{S}'_x$ is important in driving nonlinear saturation of growth and nonlinear amplification of decay of a pre-existing positive anomaly. We find that $v'\mathcal{S}'_y$ aligns in phase with $u'\mathcal{S}'_x$ during instances of large negative advection that have nonlinear contributions, irrespective of the sign of the pre-existing anomaly; and it aligns with the phase of $\delta\mathcal{S}_{QL}$ during instances of large positive advection, nonlinearly amplifying both the growth of a pre-existing positive anomaly and the decay of a pre-existing negative anomaly. It is also found that $w'\mathcal{S}'_z$ is a strong driver of amplification and decay of negative anomalies, with a larger influence in causing decay as seen by its positively skewed distribution.

6. We identify different regimes of interaction between quasilinear and nonlinear advection terms, and show that each corresponds to distinct states of the pre-existing anomaly and flow patterns related to the phasing and location of upper-tropospheric eddies. We demonstrate that large $\delta\mathcal{S}$ can not only be driven by large quasilinear advection (as has been observed before), but also by large nonlinear advection when quasilinear advection is small. On the other hand, large values of $\delta\mathcal{S}$ can be driven purely by quasilinear advection when the individual nonlinear terms are large in magnitude but opposite in phase, yielding a negligible net effect; such cases are also consistently mapped to eddy configurations. Thus, we go beyond suggesting the qualitative importance of nonlinear advection to temperature variability, and identify the particular nonlinear terms that are important for different flow configurations.

7. We obtain the contribution of identified regimes to different stages of the anomaly lifecycle, and find that different configurations corresponding to the same sign of extreme advection occupy different stages. Such an observation across both signs of pre-existing anomaly and both periods of analysis, March and April, strongly suggests the existence of different pathways to extreme states of low tropospheric DSE.

Thus, we clarify the relationship between the energetics and flow patterns in our region of interest. We demonstrate that our approach allows for a more nuanced picture of DSE variability (and extreme variability in particular) than can be achieved by traditional composites which are insensitive to the contributions of individual quasilinear and nonlinear terms.

The decision tree based methodology for identifying terms governing the differences in sign of advection may be used as a reference to identify the primary dynamical relationships interacting with a region, and the regression based analysis for identifying regions in the distribution that obey different or supplementary sets of dynamics. While we have used this methodology to understand the distribution of $\delta S$, we expect this approach should be useful in any such analysis of Reynolds-decomposed conservation laws, and could potentially also include the contributions due to the diabatic terms, if available.

Understanding the flow configurations that constitute the tail of $\delta S$ might be used to identify contextual variables governing the interrelationships between terms of advection and provide a concise summary of the relationship between the statistics and dynamics related to an observable. In the case of balanced large scale dynamics, our framework for identifying circulation patterns can help examine the structures of upper tropospheric flows interacting with a region.

Combined with the analysis mapping these circulations to stages of the anomaly lifecycle, our framework has the potential to identify persistent structures sustaining anomalies, and if there are distinct configurations associated with growth and decay. A similar approach was taken by Moron et al. (2010), who characterized distinct "weather types" associated with rainfall anomalies. However, there is still some work needed for using such a framework for relating circulations to $T_{2\,\mathrm{m}}$ variability, which is affected by other processes as previously noted.

In our case, we have also observed large durations of heatwaves associated with balanced dynamics during May, Jun, which will be studied using this framework in future work. Another useful practical application might be to explain periods of large observed skewness using such a framework. Further, the phenomenology of circulation structures obtained by studying the PDF of advection in this manner can be used towards a circulation-oriented evaluation of climate models and seasonal forecasts.

## Appendix A: Nonlinear drivers of growth and decay of a pre-existing anomaly: April

### A1 Growth of positive anomaly

We analyze the nonlinear terms driving the amplification and saturation phases of growth of a pre-existing positive anomaly (as defined in Sect. 5.1.1) during April by inspecting the nonlinear-quasilinear relationship in each of the three directions (Fig. A1) in the context of the phase relationships between $\delta S_{\mathrm{QL}}$ and $\delta S_{\mathrm{NL}}$, as follows:

- Growth of a pre-existing positive anomaly is strongly associated with positive quasilinear advective convergence (Fig. A1a, b and c). So, the problem reduces to inspecting the nonlinear drivers of the magnitude of $\delta S_{\mathrm{NL}}$ for each sign of its value.

- Figure A1a, b and c suggest that for large magnitude of $\delta S_{\mathrm{QL}} > 0$, $v'S'_y$ is consistently positive, while $u'S'_x$ is consistently negative, and $w'S'_z$ is positive but of low magnitude. Figure A1d and e clarify that the opposite phases of $v'S'_y$ and $u'S'_x$ do not overlap, since $\delta S_{\mathrm{NL}} > 0$ when $v'S'_y \gg 0$, and $\delta S_{\mathrm{NL}} < 0$ when $u'S'_x \ll 0$.

- *Nonlinear Amplification*: When $\delta S_{\mathrm{NL}} > 0$, all three nonlinear terms act in phase with it. However, $v'S'_y$ has the largest magnitude during most instances and likely drives $\delta S_{\mathrm{NL}} > 0$ when $\delta S > 0$ and $\delta S_{\mathrm{QL}} > 0$.

- *Nonlinear Saturation*: When $\delta S_{\mathrm{NL}} < 0$, the largest nonlinear contributions come from $u'S'_x$, and coincide with $\delta S_{\mathrm{QL}} > 0$. This identifies $u'S'_x$ as the driver of saturation of growth of a pre-existing positive anomaly.

### A2 Decay of positive anomaly

Using a similar approach, we analyze conditions associated with the decay of a pre-existing positive anomaly (Fig. A2).

- Decay of a pre-existing positive anomaly can be associated with positive as well as negative quasilinear advective convergence (Fig. A2a, b and c).

- *Nonlinear Amplification*: $u'S'_x$ amplifies decay by acting in phase with $\delta S_{\mathrm{QL}} < 0$ (Fig. A2e).

- Focussing on the days corresponding to $\delta S_{\mathrm{QL}} > 0$, large magnitude of $v'S'_y < 0$ (Fig. A2d) independently drives nonlinear decay.

- *Nonlinear Saturation*: Here, we focus on the days corresponding to $\delta S_{\mathrm{QL}} < 0$. $u'S'_x$, $v'S'_y$, and $w'S'_z$ act in phase ($> 0$) to drive saturation of decay by opposing the phase of $\delta S_{\mathrm{QL}} < 0$ with $\delta S_{\mathrm{NL}} > 0$ (Fig. A2d, e, f). The magnitude of $v'S'_y$ is largest, and is the likely driver of saturation of decay of a pre-existing positive anomaly.

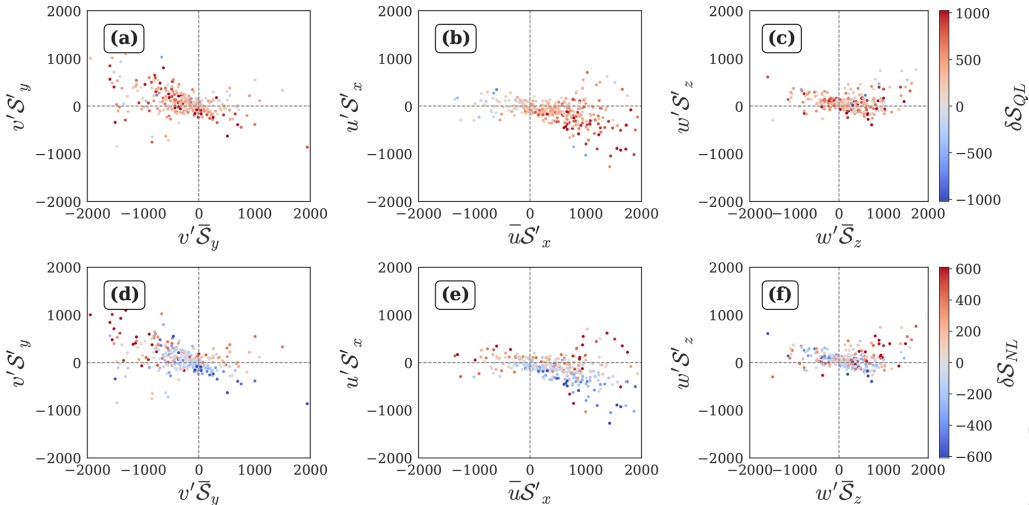

**Figure A1.** Nonlinear-quasilinear relationships for $\mathcal{S}'_{\text{Tot, Lag1}} > 0$ conditions, filtered for $\delta\mathcal{S} > 0$ days during April. **(a–c)** Quantiles of $\delta\mathcal{S}_{\text{QL}}$, **(d–f)** quantiles of $\delta\mathcal{S}_{\text{NL}}$, with blue for negative values and red for positive values and stronger colors indicating larger magnitudes. All quantities are expressed in J kg$^{-1}$.

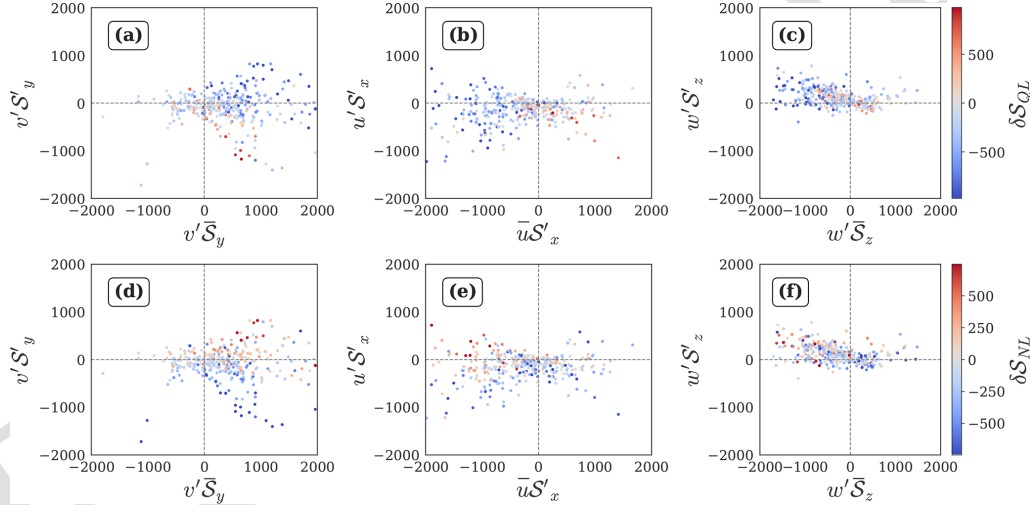

**Figure A2.** Nonlinear-quasilinear relationships for $\mathcal{S}'_{\text{Tot, Lag1}} > 0$ conditions, filtered for $\delta\mathcal{S} < 0$ days during April. **(a–c)** Quantiles of $\delta\mathcal{S}_{\text{QL}}$, **(d–f)** quantiles of $\delta\mathcal{S}_{\text{NL}}$, with blue for negative values and red for positive values, and stronger colors indicating larger magnitudes. All quantities are expressed in J kg$^{-1}$.

## A3 Growth of negative anomaly

We analyze the potential nonlinear advection terms driving the amplification and saturation phases of growth of a pre-existing negative anomaly, as follows.

5 – Growth of the pre-existing negative anomaly is strongly associated with negative quasilinear advective convergence barring a few low magnitude positive instances (Fig. A3a, b and c). So, the problem reduces to inspecting the nonlinear drivers of magnitude of $\delta\mathcal{S}_{\text{NL}}$ for each
10 sign of its value.

– The meridional nonlinear term has large excursions on the positive side (Fig. A3d), while the zonal non-linear term has large excursions on the negative side (Fig. A3e), suggesting that $v'\mathcal{S}'_y$ may act to saturate, while $u'\mathcal{S}'_x$ may act to amplify the growth of the 15 anomaly.

– *Nonlinear Amplification*: For $\delta\mathcal{S}_{\text{NL}} < 0$, both the horizontal terms are in phase, but $u'\mathcal{S}'_x$ is consistently larger in magnitude (Fig. A3e, f), affirming its role in amplifying growth. 20

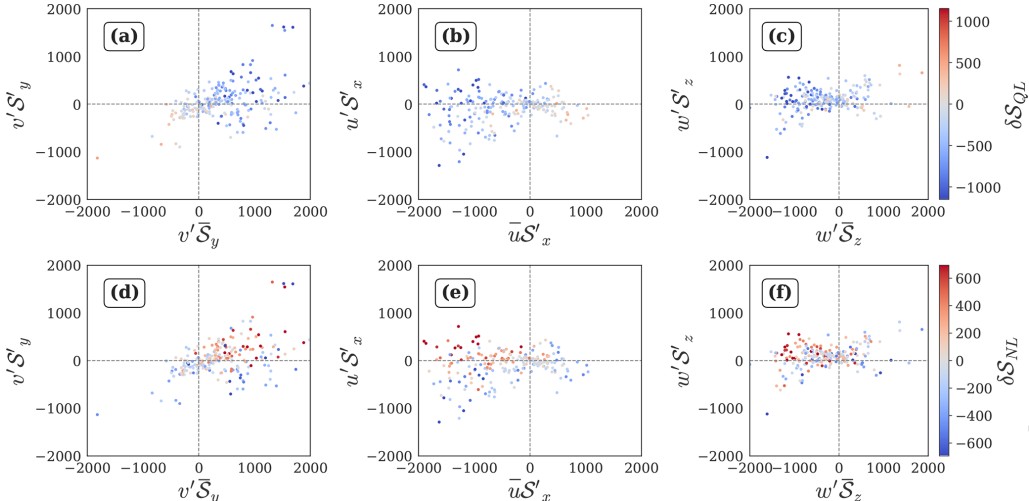

**Figure A3.** Nonlinear-quasilinear relationships for $\mathcal{S}'_{\text{Tot, Lag1}} < 0$ conditions, filtered for $\delta\mathcal{S} < 0$ days during April. **(a–c)** Quantiles of $\delta\mathcal{S}_{\text{QL}}$, **(d–f)** quantiles of $\delta\mathcal{S}_{\text{NL}}$, with blue for negative values and red for positive values, and stronger colors indicating larger magnitudes. All quantities are expressed in $\text{J kg}^{-1}$.

- – *Nonlinear Saturation*: For $\delta\mathcal{S}_{\text{NL}} > 0$, both the horizontal nonlinear terms are predominantly positive. However, $v'\mathcal{S}'_y$ shows a consistently larger magnitude than $u'\mathcal{S}'_x$ (joint distribution not shown), affirming the role of $v'\mathcal{S}'_y$ in saturating growth of the negative anomaly.

- – $w'\mathcal{S}'_z$ exhibits lower activity as compared to the horizontal nonlinear terms, indicating its limited influence in the growth phase.

## A4   Decay of negative anomaly

Next, we analyze the nonlinear advection terms associated with the decay of a negative pre-existing anomaly (Fig. A4).

- – Decay of a pre-existing negative anomaly is strongly associated with positive quasilinear advective convergence barring a few low magnitude negative instances (Fig. A4a, b and c). So, the problem reduces to inspecting the nonlinear drivers of magnitude of $\delta\mathcal{S}_{\text{NL}}$ for each sign of its value.

- – $v'\mathcal{S}'_y$ makes large contributions to both sides of zero, but its negative excursions align with large positive values of $u'\mathcal{S}'x$ or $w'\mathcal{S}'_z$. This reduces its independent role in saturation.

- – *Nonlinear Amplification*: $w'\mathcal{S}'_z$ is strongly associated with large instances of $\delta\mathcal{S}_{\text{NL}} > 0$ (Fig. A4f), corroborating the observations in Fig. 4d.

- – *Nonlinear Saturation*: $u'\mathcal{S}'_x$ is consistently of large magnitude when $\delta\mathcal{S}_{\text{NL}} < 0$ and $\delta\mathcal{S}_{\text{QL}} > 0$ (Fig. A4e), emerging as a strong driver of saturation of decay of a pre-existing negative anomaly.

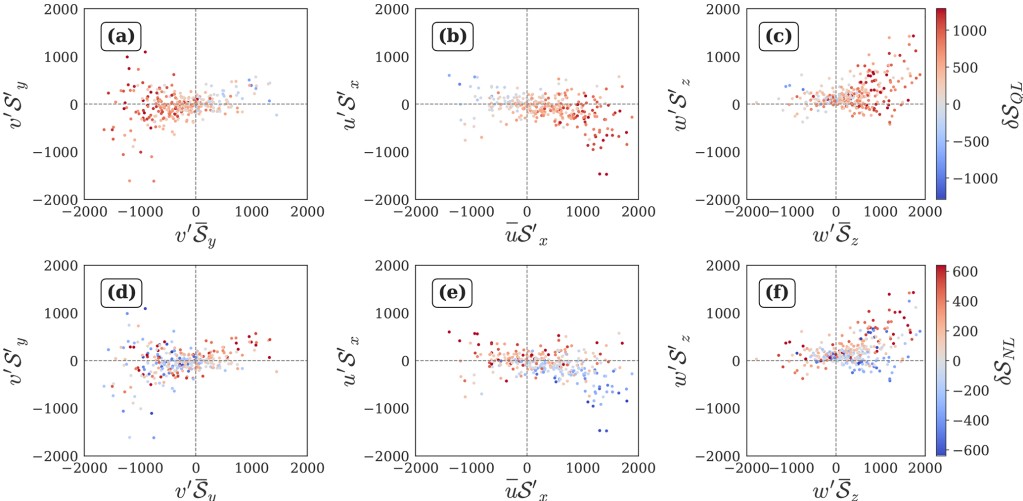

**Figure A4.** Nonlinear-quasilinear relationships for $\mathcal{S}'_{\text{Tot, Lag1}} < 0$ conditions, filtered for $\delta\mathcal{S} > 0$ days during April. **(a–c)** Quantiles of $\delta\mathcal{S}_{\text{QL}}$, **(d–f)** quantiles of $\delta\mathcal{S}_{\text{NL}}$, with blue for negative values and red for positive values, and stronger colors indicating larger magnitudes. All quantities are expressed in $\text{J kg}^{-1}$.

## Appendix B: Advection regimes: March

We repeated the analysis for March and identified the advection regimes for Growth and Decay of positive lower tropospheric DSE anomalies in Fig. B1, and for negative lower tropospheric DSE anomalies in Fig. B2, respectively.

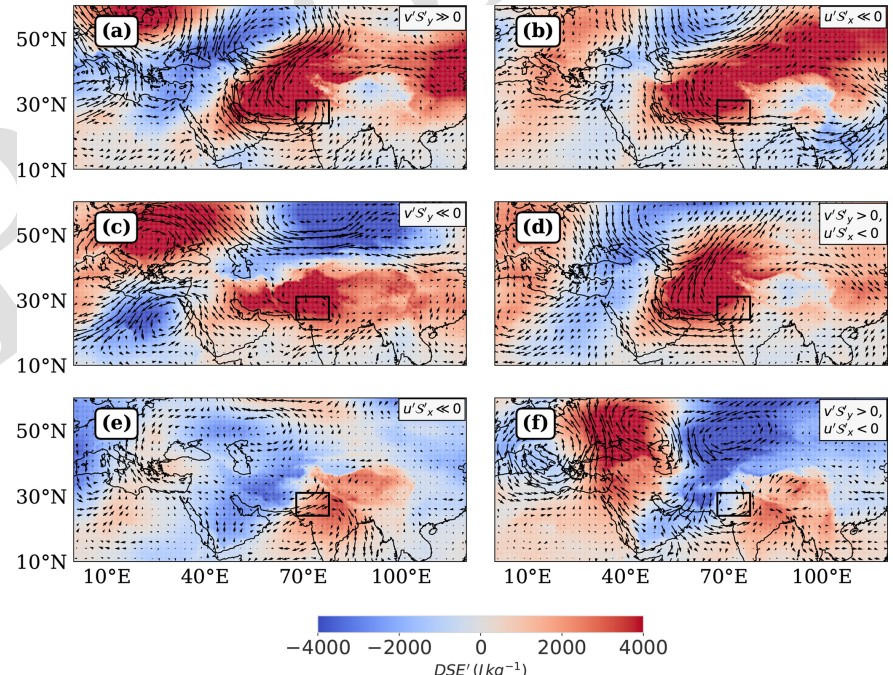

**Figure B1.** Composite representations of horizontal wind anomaly vectors and $\mathcal{S}'_{\text{Tot}}$ corresponding to advection regimes associated with Growth and Decay of $\mathcal{S}'_{\text{Tot, Lag1}} > 0$ conditions during March. The color coding represents $\mathcal{S}'_{\text{Tot}}$ values ranging from $-4000$ to $4000 \text{ J kg}^{-1}$.

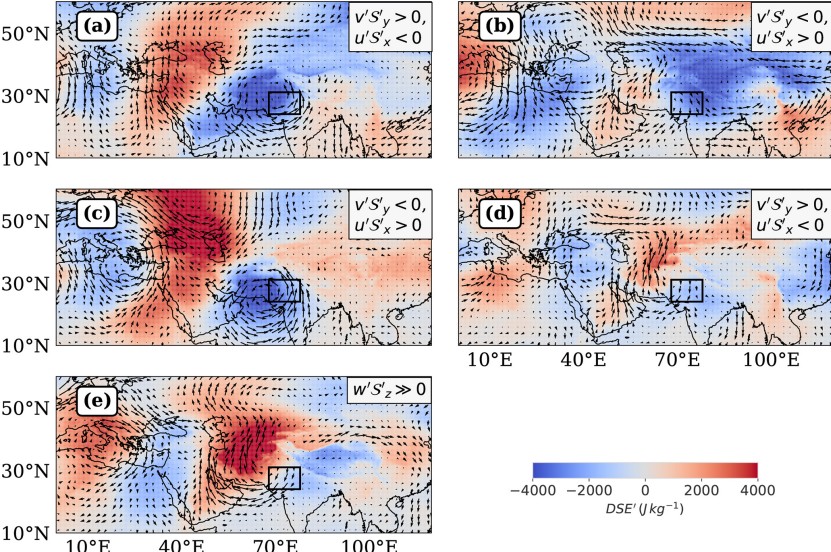

**Figure B2.** Composite representations of horizontal wind anomaly vectors and mass-weighted lower tropospheric $\text{DSE}'$ corresponding to advection regimes amplifying and dissipating $\mathcal{S}'_{\text{Tot, Lag1}} < 0$ conditions during March. The color coding represents $\mathcal{S}'_{\text{Tot}}$ values ranging from $-4000$ to $4000\,\text{J}\,\text{kg}^{-1}$.

*Code availability.* The code used for producing the figures, along with preprocessing routines, diagnostics, machine learning models and data visualizations, is archived at https://doi.org/10.5281/zenodo.17731777 TS10 (Hardik, 2025).

*Data availability.* All the data used is publicly available (ERA5; Hersbach et al., 2020).

*Supplement.* The supplement related to this article is available online at [the link will be implemented upon publication].

*Author contributions.* Both authors jointly conceptualized the project and contributed to the development of the methodology. H.M.S. developed the computer code used for data preparation and analysis, applied statistical techniques for analysis and modeling, prepared data visualization and presentation ideas, and the original draft. J.M.M. acquired the funding for this project, acted as the project administrator and supervisor, arranged for the computing resources, provided mentorship, and helped with review and revision of the manuscript.

*Competing interests.* The contact author has declared that neither of the authors has any competing interests.

*Acknowledgements.* This work was supported by the Department of Science and Technology grant DST/INT/ISR/P-40/2023 and Ministry of Earth Sciences Monsoon Mission 3 grant IITM/MM-III/2023/IND-4. We thank Vishal Dixit and Nili Harnik for helpful discussions shaping our methodology and interpretation.

*Financial support.* This research has been supported by the Department of Science and Technology, Ministry of Science and Technology, India (grant no. DST/INT/ISR/P-40/2023) and the Ministry of Earth Sciences (grant no. IITM/MM-III/ 2023/IND-4).

*Review statement.* This paper was edited by Martin Singh and reviewed by two anonymous referees.

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

**Remarks from the typesetter**

TS1    For future proofreadings, please add the entire example sentence additionally to the page and line numbers.

TS2    Please check throughout the text that all vectors are denoted by bold italics and matrices by bold roman.

TS3    Please give an explanation of why this needs to be changed. We have to ask the handling editor for approval. Thanks.

TS4    Please give an explanation of why this needs to be changed. We have to ask the handling editor for approval. Thanks.

TS5    Please give an explanation of why this needs to be changed. We have to ask the handling editor for approval. Thanks.

TS6    "th" is not superscripted according to our standards.

TS7    Please give an explanation of why this needs to be changed. We have to ask the handling editor for approval. Thanks.

TS8    Please give an explanation of why this needs to be changed. We have to ask the handling editor for approval. Thanks.

TS9    Please give an explanation of why this needs to be changed. We have to ask the handling editor for approval. Thanks.

TS10    Please confirm added DOI.