# Peer review of "The role of synoptic circulations in lower-tropospheric dry static energy variability over a South Asian heatwave hotspot"

_EGUsphere, 2025_

## Author Response (AR2)

**Author Comment**

**Hardik Shah**

**October 16, 2025**

**Response:** We thank the editor and both the reviewers for giving us an opportunity to submit a revised draft of the manuscript titled "Role of large scale circulations in driving DSE variability". We are grateful to the reviewers for the time and effort dedicated towards providing insightful comments and valuable suggestions. In addition to some of the changes suggested by reviewers, we have

- Made clarificatory changes to the Methods and Summary sections

- Changed the name of the "Efficacy" section to "Explainability and residual bias" because it seemed vague

- Revised the figure captions to provide more descriptive context, ensuring that each figure can be understood independently of the main text.

- Annotated the scatterplots in Figs. 5c, 5d in the revised manuscript to clarify the phase relationships between the nonlinear and quasilinear advection terms in the meridional and vertical directions

- Added cartoon schematics (Fig. 3 and Fig. 6 in the revised manuscript) which convey the essence of some key processes and ideas. The first schematic (Fig. 3 in the revised manuscript) describes the physical basis (in terms of the spatial structure and phase of DSE anomalies and winds) for the trivariate relationships observed in Fig. 2 and Fig. 3 of the original manuscript. The second schematic (Fig. 6 in the revised manuscript) provides physical context to the phase relationships observed in the scatterplots in Figs. 5b, 5c and 5d in the original manuscript.

- Within the "Secondary advective contributions to the $\delta\mathcal{S}$ budget" section, combined the NL-QL interactions and the Advective regimes subsections because we felt it made the presentation continuous and easier to navigate than before. Now we set the stage with a single "Advective regimes" section which explains how the NL-QL phase plots (e.g., Fig. 7 in the revised manuscript) help with the identification of regimes that actually translate to circulation patterns (e.g., Fig. 8 in the revised manuscript). We have promoted the paragraphs "Positive background conditions" and

"Negative background conditions" to the subsections "Pre-existing positive anomaly" and "Pre-existing negative anomaly", and changed the "background conditions" terminology to "pre-existing anomaly" to avoid confusion with climatological conditions.

- Added descriptions for the advection regimes in terms of the joint distribution of $\delta\mathcal{S}_{QL}$ and $\delta\mathcal{S}_{NL}$. We have also clarified that a given advection regime can be composed of multiple nonlinear combinations, and that we plot composite maps corresponding to the largest frequency combination. We have annotated all the dominant combinations in Figs. 7 and 9 in the revised manuscript and added Supplementary tables S4 and S5 that show the frequency distribution of all the nonlinear combinations that exist within an advection regime. We have added labels to the composite maps conveying the dominant nonlinear combination in each advection regime in the Figs. 8 and 10 in the revised manuscript to make it easier to differentiate between the composites. We have also made the eddy wind fields coarser to accentuate the circulation differences between the composites.

- We would like to clarify that the maps in Figs. 8 and 9 in the original manuscript had been plotted for the nonlinear combinations with the largest frequency within an advection regime, except Figs. 8a and 8f. The QL + NL Growth regime was composed of two distinct sets of nonlinear combinations, $u'\mathcal{S}_x' < 0, v'\mathcal{S}_y' < 0$ and $u'\mathcal{S}_x' \ll 0$ with sample size (n) 6 and 9, respectively. It was our oversight to annotate Fig. 7 with the $u'\mathcal{S}_x' < 0, v'\mathcal{S}_y' < 0$ combination which has the smaller sample size, as well as use it for the composite map in Fig. 8a. We have updated Fig. 9 with annotations for both the combinations making up the regime, and revised Fig. 10a in the revised manuscript with a composite map corresponding to the $u'\mathcal{S}_x' \ll 0$ instances. The QL Decay regime in Fig. 7 in the original manuscript was annotated with $v'\mathcal{S}_y' > 0, u'\mathcal{S}_x' < 0$. While it was the second largest (n=7) combination corresponding to the QL Decay regime, the largest frequency (n=12) nonlinear combination corresponded to $v'\mathcal{S}_y' < 0, u'\mathcal{S}_x' < 0, w'\mathcal{S}_z' < 0$. We had ignored this combination because it was not easy to capture the role of the vertical nonlinear term through 2D maps. However, we have decided to replace the map in Fig. 8f in the original manuscript with this combination (Fig. 10f in the revised manuscript), to maintain consistency across regimes by plotting the largest frequency combination. The conditional probability distribution tables in Figs. 8f and 9g summarized only the dominant nonlinear combinations representing each advection regime. In the revised manuscript, we have updated those tables to include all the regime samples, now presented as independent Tables 2 and 3. Accordingly, we have revised the interpretation of those tables in lines 401-408 and 474-481.

- Added a "NL saturated Decay" regime to the "Preexisting positive anomaly" section (Figs. 7, 8e in the revised manuscript).

- Edited the QL Growth regime description in the "Preexisting negative anomaly" section which was slightly inaccurate in the first draft (westerly winds were mistakenly called easterly, and the sentence construction was modified for clarity).

- Edited the 3rd Summary point to categorize the phase preferences of each of the horizontal nonlinear terms with respect to total advection and the sign of the pre-existing $DSE$ anomaly.

We would like to clarify that we had not added a composite map (in Fig. 8 in the original manuscript) for the "NL Decay ($v'\mathcal{S}'_y \ll 0$)" regime (annotated in Fig. 6 in the original manuscript), as the prominent decay regime was given by "NL Decay ($u'\mathcal{S}'_x \ll 0$)" in the same figure. We continue to preserve the annotation and do not show a corresponding composite map in Fig. 8 in the revised manuscript.

Following is a point-by-point response to the reviewers' comments and concerns.

**1 RC1**

**1.1 General Comments**

This is a well-structured preprint that presents a detailed and thoughtful analysis of the role of synoptic circulations in shaping dry static energy (DSE) variability over a South Asian heatwave hotspot. I particularly appreciated Figures 8 and 9, which offer a clear and compelling visualisation of the circulation regimes linked to different phases of DSE evolution. These figures, together with the accompanying explanations, effectively translate complex dynamical interactions into an intuitive framework. The numbered summary of key findings is also well-executed and helps ground the reader in the main contributions of the study. One area for improvement lies in the connection between the paper's motivation and its core analysis. The introduction sets up an expectation that the study will explore links between circulation and the distribution of near-surface temperature, particularly the behavior in the tails. However, the main focus is on the daily tendency of lower-tropospheric DSE ($\delta S$), without explicitly returning to the temperature distribution itself. While $\delta S$ is a justified and meaningful proxy, this disconnect could be addressed either by more explicitly linking $\delta S$ back to the statistical behavior of near-surface temperature. Even if this is just elaborating on the different stages in a lifecycle of a DSE anomaly hinted at in Fig 8f and 9g.

**Response:** We thank the reviewer for their kind words and are delighted to read their encouraging overview of our study. We appreciate your point about the introduction focussed on the tails, and the suggestion for linking

the statistics of the daily tendency of lower tropospheric DSE back to near-surface temperature. We have changed the first line of the introduction to focus on the total PDF and not just the tails, and added a quantile-quantile plot (Figure 1 in this document). We have added the conditional probability distribution of pentiles of $\delta T'_{2m}$ given the pentile of $\delta S$ in Fig. 1c in the revised manuscript for showing the distributional similarities between daily changes in near-surface temperature anomaly, $\delta T'_{2m}$, and daily lower tropospheric (600-900 hPa) advection of DSE, $\delta S$.

[Figure]

Figure 1: Quantile-quantile plots comparing the distributions of standardized transformations $(X_{std} = (X - \bar{X})/\sigma(X))$ of a) daily changes in 2m-temperature and daily lower tropospheric advection of DSE, namely $\delta T_{2m,std}$ and $\delta S_{std}$ respectively, and, b) $\delta T_{2m,std}$ and daily changes in lower tropospheric DSE, namely $\delta S_{Tot,std}$.

**1.2 Specific Comments**

1. Is there a reason why Fig 8f does not have a row for NL Growth (listed on line 431)?

   **Response:** We had omitted the NL Growth regime from the table in Fig. 8f in the original manuscript because it was seen to be active in both the neutral and the extreme deciles of $S'_{Lag1}$, and combined with its small sample size of 8, the bimodality did not seem to represent a general trend in the lifecycle of $S'_{Lag1}$. It should be noted that all the advection regimes described in Figs. 8 and 9 in the original manuscript largely represent subsets of the extreme deciles of $\delta S$, and so the regimes are made up of small sample sizes $\sim 10 - 20$ – In our period of observation, an extreme decile of $\delta S$ corresponds to a total of $30 * 43/10 = 129$ samples. We have added a row for the NL Growth regime and moved the tables

in Figs. 8f and 9g to Table 2 and Table 3, respectively, in the revised manuscript, with sample sizes mentioned in the figure caption. We note that since we are focusing on particular physical configurations rather than a statistical analysis, the sample sizes in themselves are not an issue, but the bimodality was. But in line with the reviewer's comments, we now show the NL growth regime as well since it makes the analysis complete. We plan to see if such bimodality exists when we apply our framework to a long run of a climate model (for example, using CESM LENS – `https://www.cesm.ucar.edu/community-projects/lens`).

2. `On Line 253, says similar results for both March and April. Maybe to cut down on figures could present entire analysis for just April i.e. no Figure 2, or combined March and April as done in Figure 1.`

   **Response:** We agree that it makes sense to combine the two trees in the "primary contributions" section, especially because the model results are very similar for both. We have added a combined model to Fig. 2 in the revised manuscript.

3. `I think it would be clearer to explitly define what you mean by a quasi-linear vs non-linear contribution to dS. Maybe in Fig S1, you could colour the contributions based on which category they belong to.`

   **Response:** Thanks for the suggestion. We agree and think that the tone for the rest of the paper would be appropriately set early on by coloring the bars in Fig. S1 in the original manuscript by the categories, quasilinear and nonlinear. The figure has been moved to Fig. S6 in the revised manuscript.

4. `I think there the amount of plots in figure 4 could be reduced. Possibly just showing Figure 4e, but with switch x and y axis, and then have vertical lines or shading to highlight decile 1 and 10.`

   **Response:** Thanks for the suggestion. We agree that the number of plots in Fig. 4 of the original manuscript can be reduced. Figs. 4c and 4d were only meant to accompany Figs. 4a and 4b, which are essential to identifying the regions of activity of the quasilinear and nonlinear fluxes. We have removed Figs. 4c and 4d, and the original Figs. 4e and 4f have been repositioned and renumbered as Figs. 4c and 4d in the revised manuscript.

   As per your suggestion, we have added vertical lines demarcating the extreme deciles in Fig. 4e. We think it makes sense to plot $\delta\mathcal{S}$ on the y-axis and $\delta\mathcal{S}_{QL}$ on the x-axis because Fig. 4a shows $\mathcal{S}_{QL}$ to be a strong first-order predictor of $\delta\mathcal{S}$.

   The reason for coloring the plot by $\delta\mathcal{S}_{NL}$ is to show how $\delta\mathcal{S}_{NL}$ modulates $\delta\mathcal{S}_{QL}$ to result in $\delta\mathcal{S}$. We have shortened the colorbar to make the figure

more easily consumable. We have also decided to retain Fig. 4f because it provides a clear view on the variance of nonlinear advection in each tail of $\delta\mathcal{S}$, and also validates the results in Fig. 4a.

5. `I think the supplementary information would be easier to read if all the information required for a particular figure or was given in the figure caption, rather than giving the information first and then showing the figures.`

   **Response:** We agree with the suggestion and have reconfigured the Supplementary Information document to have self-sufficient figure captions. We have added a section separately for the scale analysis of fundamental quantities.

6. `To make clear the results of Figure 8, It could be interesting to show a schematic of the growth of a given heatwave event, and the expected growth or decay regimes at different stages of its lifetime based on Fig 8f.  Or someway to make Fig 8f and 9g clearer. I find the shading in Fig 8f and Fig 9f a bit confusing, maybe make Fig 9f blue and darker blue means more negative S'.`

   **Response:**

   We wish to clarify that we are studying what casuses large changes in $DSE$ or correspondingly $T_{2m}$, and not the total value of $DSE$ or $T_{2m}$. Hence, our results do not directly correspond to growth or decay of heatwaves. However, it's a good suggestion and we thank the reviewer for the same. It is indeed our plan to use this framework for studying heatwaves, towards which we have already done some preliminary work.

   We apologize for the confusion regarding the shading in Figs. 8f and 9g in the original manuscript; the cell values were meant to reflect the distribution of regime occurrence, i.e., each cell value represented the percentage of total instances of a given advection regime that occurred during a certain background state (decile of $\mathcal{S}'_{Lag1}$). The color scheme was based in "red" for reflecting the positive definiteness of the percentage values in both the tables 8f and 9g. We have removed the color scheme and moved the tables to Tables 2 and 3 in the revised manuscript.

**1.3   Technical Corrections**

1. `Red is negative in Figures 3a and 4a but positive in Figures 2b, 3b, 5a,b,c,d.  I think it would make sense to have blue negative, grey neutral and red positive everywhere.`

   **Response:** Sorry about the miss.  We have made the color schemes consistent in the revised manuscript.

2. `In Figures 2,3 and 5 I think it may be clearer to just number the leaf nodes.`

**Response:** Thanks for the suggestion. We have retained only the leaf node number labels for both the decision trees (Figs. 2a and 5a in the revised manuscript).

3. `Fig S7 - No axis labels`

   **Response:** Sorry for the oversight. We have added axis labels and moved Supplementary Fig. S7 to Supplementary Fig. S4 in the revised Supplementary Information file.

4. `Fig S9 - Might be interesting to show decile 1 and 10 in different`
   `colour, stacked on top of each other.`

   **Response:** We agree it sounds like an interesting idea. We have moved the Supplementary Fig. S9 in the original manuscript to Fig. S11 in the revised manuscript and have plotted stacked bars with different colors for deciles 1 and 10 of $\delta\mathcal{S}$.

5. `I would be easier to read if the figures were mentioned in the`
   `text in the order that they are shown.  At the moment, the first`
   `figure mentioned is Fig 2 and the first supplementary figure mentioned`
   `is Fig S5.`

   **Response:** Thanks for pointing this out. We have reordered the supplementary figures and references in the main text to follow the order in which figures are shown.

6. `Typos`

   - `Line 253 - preenting`
   - `Line 180 - Supplimentary`
   - `Line 313 - secondory`

   **Response:** Thank you. We have made the corrections.

**2   RC2**

**2.1   Major Comments**

1. `This draft assumes that the day-to-day surface temperature variations`
   `are dominated by DSE advection processes.  It is true that Figure`
   `1 indicates a strong correlation between them, but the correlation`
   `does not show the dominance of DSE advection processes.  Imagine`
   `a case where DSE advection processes always contribute to 20%`
   `of temperature variations, they would still have a positive correlation,`
   `but the temperature variations are dominated by other processes.`
   `The authors should quantify how much DSE advection processes contribute`
   `to temperature variations -- for example, convert the x-axis in`

Figure 1(a) to a corresponding temperature variation and show all points are close to the 1:1 line.

**Response:** At the outset, we would like to clarify that day-day surface temperature variations were highly correlated with convergence of DSE advection in the *lower troposphere* (600-900 hPa) and not *near the surface* (950 hPa). Thus, while there may be diabatic effects at play near the surface, the large correlation with lower tropospheric DSE changes allows us to utilize 600-900 hPa DSE as a quantity that is highly associated with near-surface temperature and is strongly influenced by the atmospheric circulation, as has been mentioned in line numbers 101-104 of the original manuscript.

Furthermore, we have observed that daily changes in lower tropospheric DSE closely follow changes in $C_p T$, which are an order of magnitude larger than changes in $gZ$. Thus, dividing the x-axis on Fig. 1a in the original manuscript by $C_p \sim 1000$ should yield a very similar magnitude as $\delta T_{2m}$. Indeed, when we plot lower tropospheric temperature changes against $\delta T_{2m}$ in Figure 2 in this document, it is found that all points lie close to the 1:1 line.

2. A related question: Previous work (Quan et. al., 2023, https://journals.ametsoc.org/view/journals/clim/36/15/JCLI-D-22-0556.1.xml showed that diabatic processes (e.g. latent heating related to phase change), rather than horizontal dry advection processes, are responsible for extreme lower-tropospheric DSE and heatwaves especially in coastal monsoon regions. I understand this draft mainly focuses on the role of dry advection, but more discussions and comparison with the previous work are needed to explain why dry advection plays a dominant role and diabatic processes could be ignored in the zeroth order.

**Response:** Thank you for the question. We would like to start by emphasizing the point that we are studying *daily changes* in quantities like lower tropospheric DSE and $T_{2m}$, and not the absolute values of these quantities. Indeed, as we had shown in Fig. S5 of the original manuscript, the absolute quantities have a strong association with a diabatic quantity, sensible heat flux. The same had been clarified in line numbers 178-183, 186-187 in the original manuscript. We further note that high $T_{2m}$ is strongly associated with *reduced* sensible heat flux over the region, suggesting that the diabatic fluxes are responding to the atmospheric forcing, rather than the other way around.

As mentioned in the response to the previous comment, the daily changes in lower tropospheric DSE ($\delta \mathcal{S}_{Tot}$) capture a large proportion of the variation in $\delta T_{2m}$. Further, it is known that during the time of year under consideration (Mar-Apr), the region experiences very dry conditions (indeed, the region encloses a desert), as substantiated by the annual cycle of total precipitation and total cloud cover (Figs. 3a and 3b in this document).

Based on the combination of both these points involving observation and climatology, we are able to proceed with the hypothesis that diabatic heating due to condensation and surface fluxes is likely a minor player in daily surface temperature changes during Mar-Apr, and that DSE is a good approximation for the total energy budget during this season. We have added an explanation in line numbers 158-163 in the revised manuscript.

3. I do not fully understand the motivation and advantages for using the decision tree model. By doing simple conditional mean composites based on the signs or deciles of different advection terms, one could also look at different circulation patterns across the distribution of DSE advection terms. Besides, as shown in figure 2 and 3, 25% of the incidents in node 2 and 6 are not ``negative'' or ``positive'' events, so the leaf nodes are not clean classes. I suspect composite analysis of such classes with mixed events does not yield accurate descriptions of physical processes governing positive or negative DSE variations.

**Response:** a) The decision tree model was first used for developing a "first order" understanding of the advection terms governing broad differences in the magnitude and sign of of advection. The model helped us arrive at the primary mechanism generatign large fluxes of advection, compsed of the combination of the three quasilinear terms $\bar{u}\mathcal{S}'_x$, $w'\bar{\mathcal{S}}_z$ and $v'\bar{\mathcal{S}}_y$. Following the "Primary Advective Contributions" section with the "Phenomenology" section, we explained how eddy patterns correspond to large values of the three mentioned quasilinear terms. In the next section, "Efficacy", it was shown that the sum of the nonlinear terms was active with a nonzero mean in the tails of total advection. Then, using the decision tree model for nonlinear explanability of the tails in Fig. 5a, it was shown that the role of a nonlinear term was governed by the sign of $\delta\mathcal{S}$, and the sign of the total anomaly ($\mathcal{S}'_{Lag1}$).

b) In Figs. 8 and 9, we see that a given nonlinear term can either be the solely important nonlinear component (for e.g., $v'\mathcal{S}'_y$ in Fig. 8a) or covary positively or neggatively with another nonlinear term (for e.g., $v'\mathcal{S}'_y \gg 0, u'\mathcal{S}'_x \ll 0$ in Fig. 8c). Taking mean over the right extreme decile of $v'\mathcal{S}'_y$ would mask the regime in Fig. 8c. Such multivariate clusters based on different combinations of nonlinear components active in the tails yield a one-one map of circulation patterns. Thus, it is unlikely that we would be able to identify the advective signature of circulation regimes by analyzing univariate tails.

c) The composites presented in Figs. 8 and 9 in the original manuscript do not correspond to any of the decision tree nodes but to the regimes identified and annotated in Figs. 6 and 7.

4. With many DSE advection terms (i.e., dimensions of features), one can arbitrarily control the depth of the decision tree, and somehow argue that the leaf nodes represent different classes.

Why does 5 leaf nodes behave better than 4 (a shallower tree)
or 10 (a deeper tree) in figure 5? The authors might have some
early-stop criteria to avoid over-fitting, which should be justified
explicitly.

**Response:** Thanks for the question. Yes, we had applied a number of
criteria based on the sample proportion of leaf nodes, minimum entropy
decrease threshold and cost complexity threshold.

These criteria were stringent for the primary mechanism trees (Figs. 2, 3
in the original manuscript) in order to identify the broad mechanism sep-
arating the signs of daily advection, and slightly relaxed for the secondary
mechanism tree (Fig. 5 in the original manuscript) in order to identify
the importance of different nonlinear terms in the tails of advection. The
latter objective was motivated by the fact that only the meridional non-
linear term had been studied in previous studies; and the takeaway from
the Fig. 5 model was that the horizontal nonlinear terms were active for
$\mathcal{S}' > 0$ and the vertical nonlinear term for $\mathcal{S}' < 0$. The importance of the
vertical nonlinear term for $\mathcal{S}' < 0$, and the lack thereof for $\mathcal{S}' > 0$ was
corroborated with the scatterplot in Fig. 5c which shows large magnitudes
of the vertical nonlinear term only for $\mathcal{S}' < 0$.

In the revised manuscript, we have added the early-stop criteria to the
captions of Fig. 2 (which has been updated with the combined Mar-Apr
decision tree) and Fig. 5 (the secondary tree), in addition to the value of
the F1-score which we had reported previously as well.

5. A related question: In Figure 8 I think the circulation patterns
   and DSE advection processes are almost identical between (a and
   (c, as well as between (d and (e. The authors should justify
   that they are different classes physically rather than an over-fitting
   effect.

   **Response:** Thanks. We have annotated Figs. 8a-8e and 9a-9f of the
   original manuscript with the particular nonlinear term(s) active in each
   plot, as in Figs. 6 and 7 of the original manuscript. This may make it easier
   to locate the active anomalous wind components in each plot and identify
   the differences between plots like a/c and d/e. For example, $v'\mathcal{S}'_y$ is active
   in 8a but both $v'\mathcal{S}'_y$ and $u'\mathcal{S}'_x$ are active in 8c. Figure 9a-9f in the original
   manuscript has been moved to Fig. 10 in the revised manuscript. As
   mentioned earlier in the document, the transition matrices in Figs. 8f and
   9g have been moved to Tables 2 and 3 for allowing space for annotations
   in the composite plots.

**2.2 Minor Comments**

1. Equation 1a and 1b: The authors should quantitatively justify
   that the non-divergent approximation is reasonable.

[Figure]

Figure 2: Scatterplot of daily lower tropospheric temperature changes ($\delta T_{600-900\,hPa}$) against daily 2m temperature changes $\delta T_{2m}$. The solid line has slope = 1 and passes through the origin.

[Figure]

Figure 3: Monthly mean quantities computed over the period 1980-2022 using the ERA5 reanalysis product. a) Total Precipitation as the sum of large-scale and convective precipitation, expressed as depth (in metres) the water would have if it were spread evenly over the region of interest. b) Total cloud cover expressed as the proportion of the entire atmosphere over the region of interest covered by cloud.

**Response:** Thank you for the question. While this was termed as a minor comment, we actually spent most of our time during this revision addressing this comment!

We performed a quantitative analysis and saw that the divergent term was not small (Fig. 4). However, we believe that our use of the advective term alone is still justified, for reasons we state subsequently. We start by emphasizing that neglecting the divergence term does not change any of the core results of our study, which are concerned with the identification and characterization of the energetic signature of eddy configurations. However, what might change is the association between advective fluxes and $\delta\mathcal{S}_{Tot}$ or $\delta T_{2m}$, which might be important in other studies which require closure of the DSE budget.

Firstly, our focus is primarily on large magnitude of daily changes. The reason for this focus was that low magnitude changes were not clearly associated with coherent structures. The tails of the univariate distributions of each of $\delta\mathcal{S}_{Tot}$, $\delta\mathcal{S}$ and $\delta\mathcal{T}_{2m}$ are related as seen in the q-q plots in Fig. 1. As explained in the Phenomenology section in the main text, large instances of $\delta\mathcal{S}$ are associated with such coherent configurations as large scale eddies, which makes the tails of this distribution most amenable to a dynamical analysis.

We see that the error of the nondivergent approximation is much smaller in the extreme deciles than in the body of the distribution – the 75th percentile of errors in each of the extreme deciles is below $\sim 50\%$ of $\delta\mathcal{S}_{Tot}$ as shown by the upper edge of the respective boxes, and the error is higher in the middle deciles which correspond to low magnitude data points of $\delta\mathcal{S}_{Tot}$.

The variance of errors is also much larger in the middle deciles than the extreme deciles, suggesting that while small magnitudes of DSE change might have a strong influence of divergent and/ or diabatic effects, the large magnitudes of DSE change are strongly driven by advection which corresponds to eddy activity.

Secondly, we were interested to characterize advective component more carefully as we view our work as an extension of the community's current understanding of the role of eddy-eddy interactions in shaping the tails of temperature distributions (existing literature cited in the main text, most of which has primarily analyzed the variety of advective contributions like eddy-mean and eddy-eddy interactions to temperature anomaly in the midlatitudes). However, we are coming to realise that the divergent term may be more important away from the midlatitudes (where most current literature is focused on) in our subtropical region of interest, as we have also observed in ongoing work. We intend to explore this further in future work.

Thirdly, there has been acknowledgement of uncertainty in the ERA5 divergent term due to "contamination by numerical noise" in the overview

of the following ERA5 dataset - `https://cds.climate.copernicus.eu/datasets/derived-reanalysis-energy-moisture-budget?tab=overview` - that provides monthly means of vertically integrated mass-consistent energy flux estimates. The implications of this uncertainty for our understand of the dynamics is unclear, and we consider this to be a minor factor to keep our focus on the advective term.

2. `Equation 3d only has eddy-mean terms and the eddy-eddy term. Why is the mean-mean term ignored?`

   **Response:** Sorry for missing this term in the equation. The mean-mean term was not included in equation 3d but was certainly included in the analysis. We have changed the order of the equations and added all four terms – eddy-mean, eddy-eddy, mean-eddy and mean-mean to equation 3b in the revised manuscript.

3. `Why is the eddy-mean term in equation 3d named the ``quasilinear'' term, not the linear term?`

   **Response:** We refer to the eddy–mean advection term (e.g. $v'\bar{\mathcal{S}}_y$) as "quasilinear" because it is linear in the spatial derivative of the scalar field $\bar{\mathcal{S}}$, while the coefficient of the derivative (eddy flow $v'$ in this example) varies in space and time, and may be nonlinear in the broader dynamical system from which it is derived. The nonlinear characterization of $v'$ might be especially appropriate when computed externally - for instance, from reanalysis datasets - and not solved within the same PDE framework. Although the full system may exhibit nonlinear behavior, the advection term itself remains linear in the highest-order spatial derivative, which aligns with the standard definition of quasilinear structure in PDE theory. This usage is consistent with the framework proposed by Marston et al. (2016, Phys. Rev. Lett.), who formalize quasilinear approximations by retaining linear evolution of small-scale modes while allowing nonlinear interactions with large-scale components.

4. `Line 266 and 282: Composite circulation maps, perhaps in the SI, will make the two kinds of deviations easier to understand.`

   **Response:** Thanks. We have added composite circulation maps for the two kinds of deviations to Supplementary Fig. S10.

5. Line 358: The full-stop in the middle should be removed. **Response:** Sorry for the negligence. We have made the change in the revised manuscript.

6. Line 378: These findings finding underscore ... **Response:** We have made the change in the revised manuscript.

[Figure]

Figure 4: Boxplots of error fraction associated with the advective approxima-tion of daily lower tropospheric DSE changes, $\delta\mathcal{S}_{Tot}$, grouped by decile of $\delta\mathcal{S}_{Tot}$. Error fraction is defined as the difference between daily lower tropospheric DSE changes and daily lower tropospheric DSE advection ($\delta\mathcal{S}_{Tot}$ and $\delta\mathcal{S}$, respec-tively) divided by $\delta\mathcal{S}_{Tot}$.